# Intra-promoter switch of transcription initiation sites in proliferation signaling-dependent RNA metabolism

Joseph W. Wragg [1] ✉, Paige-Louise White[1,3], Yavor Hadzhiev [1,3],
Kasun Wanigasooriya[1,2], Agata Stodolna[1], Louise Tee[1], Joao D. Barros-Silva[1],
Andrew D. Beggs [1,2] ✉ & Ferenc Müller [1] ✉

Global changes in transcriptional regulation and RNA metabolism are crucial features of cancer development. However, little is known about the role of the core promoter in defining transcript identity and post-transcriptional fates, a potentially crucial layer of transcriptional regulation in cancer. In this study, we use CAGE-seq analysis to uncover widespread use of dual-initiation promoters in which non-canonical, first-base-cytosine (C) transcription initiation occurs alongside first-base-purine initiation across 59 human cancers and healthy tissues. C-initiation is often followed by a 5′ terminal oligopyrimidine (5′TOP) sequence, dramatically increasing the range of genes potentially subjected to 5′TOP-associated post-transcriptional regulation. We show selective, dynamic switching between purine and C-initiation site usage, indicating transcription initiation-level regulation in cancers. We additionally detail global metabolic changes in C-initiation transcripts that mark differentiation status, proliferative capacity, radiosensitivity, and response to irradiation and to PI3K–Akt–mTOR and DNA damage pathway-targeted radiosensitization therapies in colorectal cancer organoids and cancer cell lines and tissues.

Transcription regulation is a defining factor in cancer development. Detection of transcript abundance is diagnostic and reveals mechanisms of malignant transformation. Transcription initiation at the core promoter reflects a fundamental regulatory level, creating dynamic transcript variation through alternative promoter usage in cancers[1,2]. It can influence all levels of RNA metabolism, functioning as a first step in regulating post-transcriptional processing and translation efficiency[3,4]. This is exemplified by the evolutionarily conserved translation machinery genes, transcribed into mRNAs with a C base at the 5′ end, followed by a terminal oligopyrimidine stretch, called 5′TOP[5]. These transcripts are distinctively regulated in cellular stresses and various cancers, predominantly through modulation by the phosphoinositide

3-kinase (PI3K)−mammalian target of rapamycin (mTOR) signaling pathway[6–11]. mTORC1 signaling phosphorylates and inactivates 4EBP1 and LARP1, both selective regulators of 5′TOP mRNA cap interaction with the eIF4E−eIF4G1 translation initiation complex. In cellular stress, mTOR signaling is lost and therefore 5′TOP mRNA translation initiation is inhibited[12–17]. However, there is a gap in knowledge of how patterns of transcription initiation and selection of start sites influence RNA metabolism and cancer progression.

We explored the role of transcription initiation dynamics within the core promoter in cancer cell identity and behavior. We uncovered a previously unappreciated dynamic of transcription initiation choice, and demonstrate distinct regulation of transcripts emanating from

[1]Institute of Cancer and Genomic Sciences, College of Medical and Dental Sciences, University of Birmingham, Birmingham, UK. [2]Present address: Department of Surgery, University Hospitals Birmingham National Health Service (NHS) Foundation Trust, Birmingham, UK. [3]These authors contributed equally: Paige-Louise White, Yavor Hadzhiev. ✉e-mail: j.wragg@bham.ac.uk; a.beggs@bham.ac.uk; f.mueller@bham.ac.uk

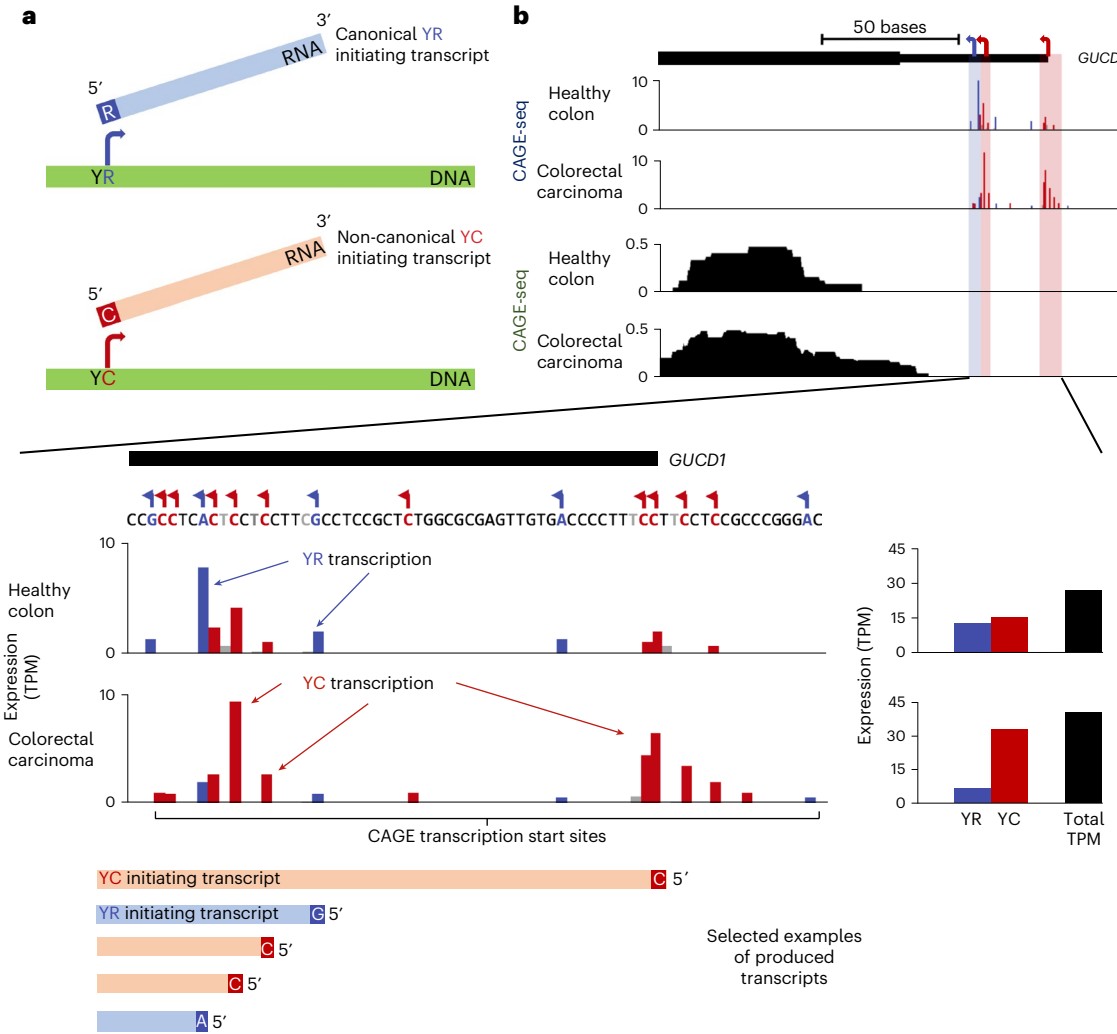

**Fig. 1 | Dual-initiation promoter usage with non-canonical YC transcription in cancer. a**, Illustration of transcription from canonical (YR) and non-canonical (YC) initiation dinucleotides. **b**, UCSC Genome Browser view of CAGE-seq and RNA-seq tracks from a representative dual-initiator gene, guanylyl cyclase domain containing 1 (*GUCD1*), where there is a switch from canonical (YR) transcription (blue bars) in the healthy tissue to non-canonical (YC) transcription (red bars) in the cancer (colorectal carcinoma). The rare instances in which transcription initiates from an alternative site besides YR and YC are shown in gray. Bar graphs to the right show total TPM values of YR and YC transcription in healthy and cancer tissues together with overall TPM values. Selected examples of transcripts produced at varying levels from the GUCD1 promoter, between the healthy and cancer samples, are illustrated below.

the same core promoter but with distinct 5′ end nucleotides. This regulation plays a role in cancer cell behavior, differentiation status and response to radiotherapy in colorectal cancer (CRC) models. We identified the PI3K–Akt–mTOR–Myc–p53 regulatory network, previously implicated in ribosome synthesis regulation (reviewed in refs. 18,19), defining alternative transcription initiation site usage.

In the majority of genes, transcription initiates from a well-characterized motif consisting of a pyrimidine (C/T, IUPAC code Y) at −1 bp upstream of the transcription start site (TSS) followed by a purine (A/G, IUPAC code R) at the +1 position. This constitutes the canonical 'YR' motif in mammals (reviewed in ref. 20) (Fig. 1a). However, a subset of predominantly protein translation machinery-associated genes initiate from an alternative motif, with cytosine at the +1 position of the TSS, called the TCT motif in *Drosophila* (Fig. 1a)[21,22] and the YC motif in vertebrates[23]. The TATA box-binding protein (TBP)-related factor TRF2 has been identified to selectively mediate the transcription of TCT promoter genes in *Drosophila*[24]; however, despite evolutionary conservation, vertebrate regulation of YC transcription initiation is largely unknown.

Recently, we challenged the exclusivity of YC-initiating 5′TOP mRNAs to translation machinery genes by demonstrating that thousands of protein-coding genes carry both YR and YC transcription

initiation, often intermingled on promoters. Genes in this group are termed dual-initiating genes and are evolutionarily conserved from *Drosophila* to vertebrates[21,23]. Intermingled R-start and C-start mRNA species can display distinct transcriptional and post-transcriptional RNA dynamics[23]. Since YC initiation-linked 5′TOP mRNAs are targets of cancer-associated transcript metabolism[3,15,25], we reasoned that pervasive use of YC initiation in a large number of genes may expand the potential for tumorigenic and prognostic transcriptional changes. Moreover, as the differential initiation dynamics are not restricted to alternative promoters[1], but mostly occur within promoters, we postulated that previously unappreciated complexity of transcript isoforms may be generated from the same genes with distinct 5′ end sequence composition and associated post-transcriptional fates in cancers.

## Results

### 5′-C transcripts enriched in poorly differentiated, proliferative cancers

Tumorigenesis is often marked by increased activity of mRNA translation machinery, encoded by transcripts bearing the 5′TOP motif and transcribed from YC initiators[6–11,26]. Since transcripts initiating from YC dinucleotides are pervasive across the genome[23], we reasoned that

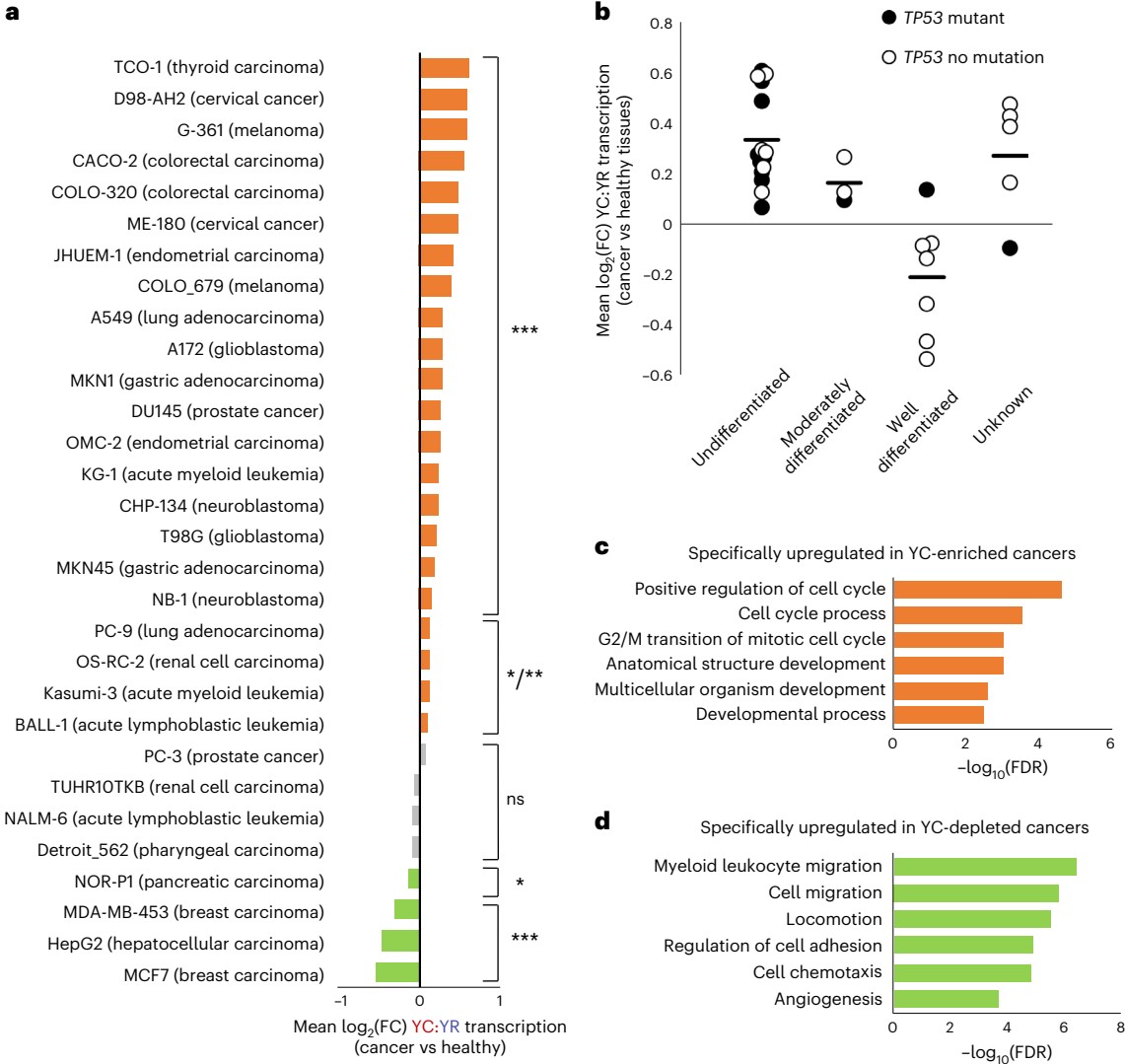

**Fig. 2 | 5′-C transcripts are most enriched in poorly differentiated and proliferative cancer types.** Dual initiators (promoters with >1 TPM for both YC and YR transcription in the majority of datasets (>30 out of 59)) were identified across all selected FANTOM5 cancer and healthy tissue CAGE datasets (*n* = 3,475). The relative expression of the YC versus YR component of transcription for each dual initiator was calculated and compared between matched cancer and healthy tissues. **a**, Table of cancer samples ordered by mean log₂ fold change (log₂(FC)) in the ratio of YC to YR transcription of dual-initiator promoters between cancer and matched healthy tissue. *P* value (paired two-tailed *t*-test) of this expression change is also shown (ns, not significant; *P < 0.05; **P < 0.01; ***P < 0.001; full list of *P* values available in the Source data). This table is color-coded to show cancers where YC transcription at dual initiators is significantly enriched (orange), unchanged (gray) or depleted (green) relative to the matched healthy tissues. **b**, The differentiation status of each cancer sample was identified where publicly available. They were then separated into undifferentiated, moderately differentiated and well-differentiated cancer types, and the distribution of each sample's mean log₂(FC) in YC:YR transcription (as calculated in **a**) was plotted for each group. The distribution of *TP53*-mutant samples is also shown by the color of each plot point. **c**,**d**, Biological process ontology of genes specifically upregulated in YC-enriched cancers (*n* = 132) (**c**) and YC-depleted cancers (*n* = 144) (**d**). FDR, false discovery rate.

5′TOP mRNAs in tumor development may be extended to YC RNAs produced in dual-initiating genes, owing to a potential shared targeting mechanism. To test this hypothesis, we measured 5′-C RNA content in various types of tumors representing different phases of tumorigenesis.

YR and YC initiation (Fig. 1a) can be distinguished by 5′ end detection at nucleotide resolution by cap analysis of gene expression sequencing (CAGE-seq)[23]. CAGE-seq detects altered transcription initiation patterns in cancer as demonstrated by the promoter of *GUCD1* (encoding guanylyl cyclase domain containing 1), a gene associated with enhanced cell proliferation and invasion of cancer cells[27,28]. CAGE-seq analysis in published FANTOM5 datasets[29,30] revealed it to be a dual-initiator promoter (DIP) in healthy colon, with both YC and YR initiation present. In CRC samples, however, the YC component of expression is dramatically enhanced at the expense of the YR

component (Fig. 1b), while overall expression is moderately changed. The dynamic shift in initiation site usage is invisible by traditional RNA-seq transcriptomics, although a slight elongation of reads mapping to the first exon occurs (Fig. 1b).

We then asked about the pattern of 5′-C transcript abundance in other DIPs by surveying FANTOM5 CAGE-seq data from 59 datasets, representing 17 cancer and matched healthy tissues (Supplementary Table 1). We identified CAGE transcription start sites (CTSS) that clustered into 17,480 consensus clusters between samples, representing promoter regions as previously described[31]. CTSS within consensus clusters were segregated into YR and YC classes with tag per million (TPM) values calculated. Dual-initiating promoters were identified (*n* = 3,475) with consensus clusters of >1 TPM for both 5′-R transcription and 5′-C transcription in the majority of datasets (30 datasets). The ratio

of YC to YR transcription was calculated for these DIPs across the datasets and compared between matched cancer and healthy samples. This analysis revealed that the majority (22 out of 30) of cancer cell types had significantly enhanced YC transcription within DIPs (YC-enriched tumors). There was considerable heterogeneity between cancer types, with four cancers with significantly depleted 5′-C transcript usage relative to their matched healthy cell types (YC-depleted tumors) and four cancers with no significant change (YC-neutral tumors; Fig. 2a).

To identify features linked to enriched YC initiation, we assessed publicly available profiling data for well-characterized tumor cell lines (Supplementary Table 2). Segregating the tumors by differentiation status showed a clear contrast in the average YC usage of DIPs between poorly differentiated (100% YC-enriched) and well-differentiated (86% YC-depleted) tumors (chi-squared test, $P < 0.001$) (Fig. 2b). Of note, tumor protein P53 (*TP53*) mutation status also trended with YC-enriched cancers, with mutation-bearing tumors representing 48%, 25% and 20% of YC-enriched, YC-neutral and YC-depleted tumors, respectively (chi-squared test, $P = 0.42$) (Fig. 2b and Supplementary Table 2). To further mutational associations, data were extracted from the Cancer Cell Line Encyclopedia[32], with coverage for 24 of 30 lines. This revealed that mutations in *KIAA0586*, *CLCN3*, *ZNF22*, *MRPS16*, *DLEU7*, *TBX2*, *MKKS*, *DVL1* and *ADGRG7* were all significantly associated with YC-depleted cancers, while *USH2A* mutations were significantly associated with YC-enriched cancers (Supplementary Table 3).

To further explore the factors segregating the YC-enriched versus YC-depleted cancer types, we identified genes with expression directly correlated with YC. We identified genes with an average of greater than twofold enrichment in expression in YC-enriched cancers versus matched healthy tissues and a greater than twofold depletion in expression in YC-depleted cancers versus matched healthy tissues (Supplementary Table 4). Almost half of these genes (22 out of 52) were associated with differentiation or stem cell character (Supplementary Table 4), suggesting a link between tumor differentiation status and YC–YR transcription initiation choice. This is exemplified by the dual-initiating oncogene *ABI1* (encoding Abl Interactor 1), a component of the WAVE complex[33]. In healthy bronchial epithelial cells, and in the well-differentiated lung cancer line (PC9), its expression was predominantly from YR initiation sites. In the undifferentiated lung cancer line (A549), however, initiation switched dramatically towards a dominant YC transcription initiation site (Extended Data Fig. 1a).

Next, we sought to identify genes specifically upregulated in YC-enriched versus YC-depleted cancers. We identified genes consistently upregulated in YC-enriched cancers over matched healthy tissues (enriched by more than twofold in >75% of samples) and unchanged or depleted in all YC-depleted cancers and vice versa. This analysis identified 132 and 144 genes specific to YC-enriched or YC-depleted cancers, respectively. Gene ontology analysis revealed cell cycle control and proliferation genes upregulated in YC-enriched cancers (Fig. 2c) in

contrast to cell migration in YC-depleted cancers (Fig. 2d). To further investigate this finding, we correlated YC dynamics with published cell doubling rate and metastatic association (Extended Data Fig. 1b,c) in cancer cell lines. There was no association with doubling rate, but a non-significant enrichment for cancers from patients with metastasis at collection in the YC-depleted cohort was found (metastasis reported in 36.3%, 0% and 75% of YC-enriched, YC-neutral and YC-depleted cancers, respectively; chi-squared test, $P = 0.16$).

We asked which DIPs showed a cancer cell type-defining switch in YC:YR TSS usage. We calculated the average YC:YR ratio for each DIP within each cohort, selecting promoters in which the YC:YR ratio dynamically changed from the highest value in YC-enriched cancers, to an intermediate value in YC-neutral cancers, to the lowest value in YC-depleted cancers. A total of 422 genes showed a cohort-dependent trajectory of YC:YR transcription initiation levels (Extended Data Fig. 1d and Supplementary Data. 1) with significant enrichment for chromatin and organelle organizational genes, in line with the proliferative activity of YC-enriched cancers, as well as genes regulated by the PI3K–Akt–mTOR and Myc regulatory axis (0.03–0.05 false discovery rate) (Extended Data Fig. 1e). Both these pathways regulate 5′TOP-containing ribosome gene transcripts and ribosome biogenesis[18,19], and our findings suggest their wider role in regulating YC:YR ratios and differential RNA metabolism from dual-initiating promoters.

Taking these findings together, a segregation of transcripts between YC-enriched and YC-depleted cancers was seen, with enrichment of YC initiation correlated with poorly differentiated, proliferative (and potentially *TP53*-mutated) cancer subtypes. Intriguingly, these factors have all been associated with radiotherapy response[34–37] and raise the potential for TSS as a prognostic indicator of tumor subtypes and therapy response.

### Enriched YC initiation marks radiotherapy-responsive CRC tumors

To investigate whether 5′-C transcript abundance influences radiotherapy response, we focused on CRC, one of the YC-enriched cancer types (Fig. 2a). We performed CAGE-seq on treatment-responsive and non-responsive CRC formalin-fixed paraffin-embedded (FFPE) pre-treatment biopsy specimens (Supplementary Table 5). Samples were selected based on the response of the donor tumors to a standard course of neoadjuvant chemo-radiotherapy (45–50 Gy over 32–39 weeks, alongside capecitabine/5-FU) with either robust tumor regression (M1) or no response (M4–M5) (Extended Data Fig. 3a and Supplementary Table 5). Four responsive and four non-responsive tumor samples were sequenced using FFPEcap-seq[38] and mapped to the GRCh38/hg38 genome. Around 55–60% of CTSS mapped to promoter regions, as expected for FFPEcap-seq[38] (Extended Data Fig. 2a and Supplementary Table 6). Biological replicates of responsive and non-responsive CRC tumors were merged, and total expression

**Fig. 3 | Enriched YC initiation marks radiotherapy-responsive CRC tumors.**
**a**, Bar graph of the total expression from all CTSS within consensus clusters, initiating with YR or YC dinucleotides, between the responsive and non-responsive CRC clinical tumor cohorts (chi-squared, $P = 0.0001$). **b,c**, Dual-initiating promoters were identified as before ($n = 186$). The proportion of transcription initiating in each dual-initiator promoter, from the YC and YR sites, was quantified for each sample and compared between them on a per-promoter basis. **b**, Frequency distribution graph showing the degree of expression change of the YC component (normalized to YR) of each dual promoter between responsive and non-responsive CRC clinical tumor samples (paired two-tailed *t*-test, $P = 0.023$). **c**, Frequency distribution graph as in **b** but with each expression component (YR and YC) separately analyzed (paired two-tailed *t*-test, *$P = 0.013$). **d**, Plot of relative survival of the five CRC organoid cultures under study, between those irradiated with 25 Gy versus 0 Gy ($n = 3$ independent experiments, data are presented as mean values ± s.e.m.). **e**, Bar graphs of the total expression from CTSS in dual-initiator consensus clusters (left, $n = 6,292$) and all other consensus clusters (right, $n = 12,428$), initiating with YR or YC dinucleotides, for

each CRC organoid sample (chi-squared test, $P < 0.0001$ for both dual and other promoters). **f**, Dual-initiator promoters were identified as before ($n = 6,292$), and the YC:YR expression ratio was calculated for each in all organoid samples and divided by the average YC:YR ratio for that promoter. The frequency distribution of these values, illustrating the YC:YR ratio of transcription for all dual initiators, between samples is shown. **g**, UCSC Genome Browser view of CAGE tracks from a representative dual-initiator gene, *SND1* (staphylococcal nuclease and tudor domain containing 1), showing a dynamic switch from YC-predominant transcription in radiotherapy-responsive CRC organoids (CRC1) to YR-predominant transcription in radiotherapy-non-responsive organoids (CRC5), with balanced transcriptional output from YR and YC components in the moderately responsive CRC organoid (CRC3). **h**, Bright-field images showing the morphology and doubling times of the five CRC organoid lines under investigation. White arrows, cysts; red arrows, crypts; scale bar, 100 μm. Doubling time analysis is based on measurements from three independent experiments.

from all CTSS with YR or YC initiation was compared between the cohorts (Fig. 3a). The YC:YR ratio was significantly ($P < 0.001$) shifted, with YC initiation enriched in tumors responsive to chemo-radiotherapy.

To understand the source of this change, we calculated the YC:YR ratio dynamics of DIPs, grouped CTSS into 3,472 consensus clusters between samples, identified 186 DIPs and compared the YC:YR ratio

of DIPs between the responsive and non-responsive cohorts (Fig. 3b). A significant ($P = 0.02$) enrichment in the YC:YR ratio was found. To determine the contribution of both types of initiation site selection, we compared 5′-C and 5′-R transcript levels separately between the responsive and non-responsive cohorts (Fig. 3c). 5′-C transcripts were specifically enriched in the responsive cohort, while the 5′-R

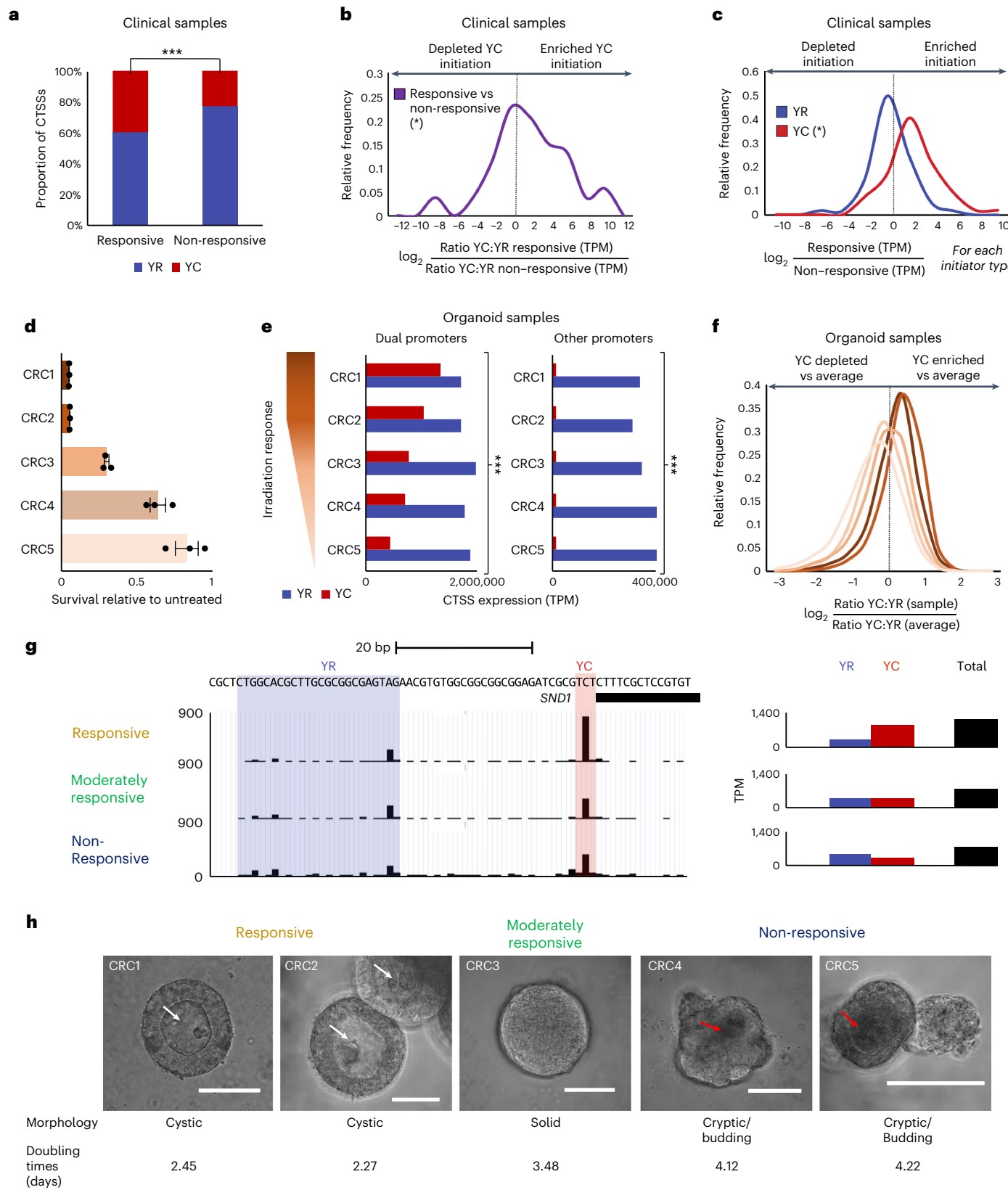

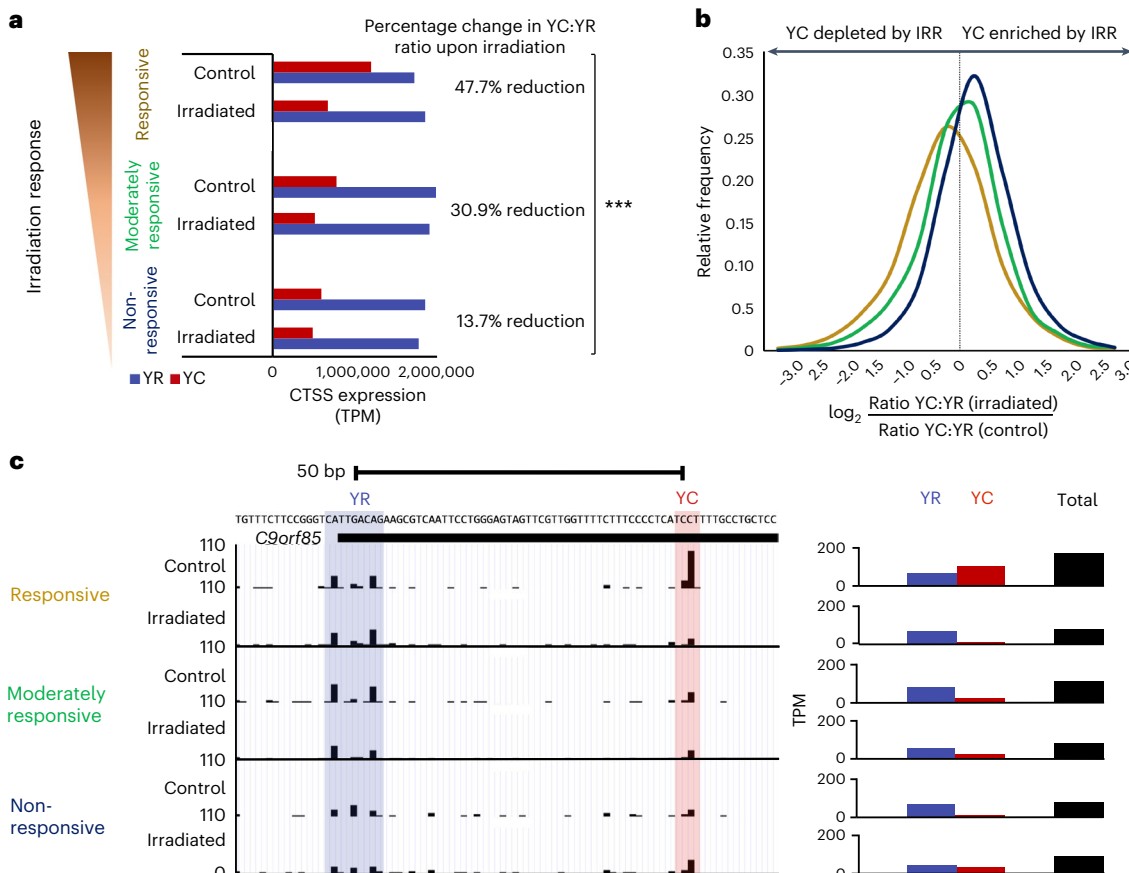

**Fig. 4 | Radiotherapy-responsive modulation of YC transcription initiation correlates with CRC clinical response. a**, Bar graph showing the proportion of CTSS in dual-initiator consensus clusters, initiating with YR or YC dinucleotides, in responsive (average of CRC1 and CRC2), moderately responsive (CRC3) and non-responsive (average of CRC4 and CRC5) organoids treated with 0 Gy (control) or 25 Gy (irradiated) irradiation treatment (chi-square analysis, $P = 0.0001$). **b**, Frequency distribution graph showing the degree of expression change of the YC component (normalized to YR) of each dual promoter upon irradiation in responsive (average of CRC1 and CRC2), moderately responsive (CRC3) and non-responsive (average of CRC4 and CRC5) organoids. **c**, UCSC Genome Browser view of an irradiation-responsive dual promoter (*C9orf85*). This promoter shows a clear loss of the YC component upon irradiation in responsive tumors (CRC1) and a relative lack of the YC component in moderately responsive (CRC3) and non-responsive (CRC5) organoid samples.

component was modestly depleted (Fig. 3c). This suggests that the YC:YR change in DIPs was due primarily to enrichment of YC initiation in chemo-radiotherapy-responsive tumors and that the total gene expression variation was due to selective differential metabolism of 5′-C transcripts.

It is possible that differences in initiation site usage were due to potentially unequal RNA degradation in FFPE archived CRC clinical samples. Additionally, FFPEcap-seq is less efficient in identifying DIPs than traditional CAGE (186 DIPs (5.3% of all promoters) versus 3,475 DIPs (19.9% of all promoters) in FANTOM5 data). Therefore, we aimed to confirm our observations on CRC organoid samples by CAGE-seq (CRC1–CRC5) (Supplementary Table 5).

We characterized the radiotherapy responsiveness of these organoids upon exposure to 25 Gy of irradiation over 5 days, followed by a 5-day recovery period to allow the physiological and transcriptomic effects of irradiation to register in the samples, mimicking the standard short-course radiotherapy protocol given clinically in rectal cancer[39,40]. This analysis revealed significant differences in the radiotherapy responsiveness between the organoid lines, with two showing a robust ~95% reduction in viability (CRC1 and CRC2) compared to untreated samples, one with moderate response (70% viability reduction, CRC3) and two with little response (36% and 17.5% viability reduction, CRC4 and CRC5, respectively) (Fig. 3d). This finding allowed us to investigate differences in promoter usage across organoids with a range of radiosensitivities.

As before, CAGE reads were mapped and CTSS assigned, with ~90% of CTSS mapping to the promoter region of genes (Extended Data Fig. 2a and Supplementary Table 6). The CTSS were clustered into 18,713 consensus clusters. Notably, this revealed that the YC:YR ratio was directly correlated with radiotherapy response, displaying a continuous gradient from 33–36% of transcripts starting with C in the most responsive organoid samples (CRC1 and CRC2) to 15% in the least responsive sample (CRC5) (Extended Data Figs. 2b and 3b), in agreement with the clinical samples (Fig. 3a). To explore whether this dynamic was a property of altered use of DIPs rather than a global transcript-level change, we identified DIPs as before ($n = 6,285$, 34% of consensus clusters) and analyzed CTSS expression from all DIPs versus all other promoters ($n = 12,428$) (Fig. 3e). This analysis revealed that (1) the radiotherapy response-associated YC:YR dynamic was significant in both dual-initiating and non-dual-initiating promoters; (2) other promoters almost exclusively generate YR transcripts, with promoters generating only YC-initiating transcripts a very rare event (six consistently across all CRC organoid samples and ~30 per sample); (3) the change in the YC:YR ratios in DIPs was predominantly due to altered 5′-C transcript levels (in agreement with Fig. 3c and Extended Data Fig. 3c); and (4) the vast majority (~83%) of transcripts in the CRC organoid samples emanated from DIPs. Further to this finding, frequency distribution analysis of YC:YR ratios within DIPs again showed enrichment in the YC content in radiotherapy-responsive samples relative to non-responsive samples

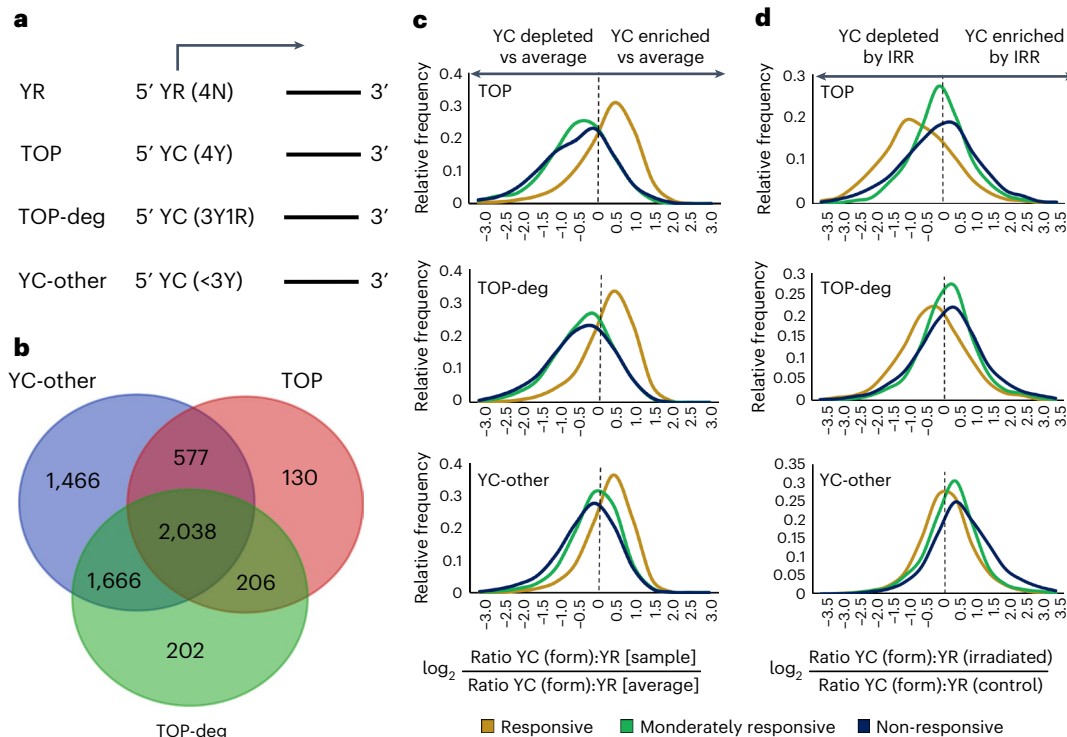

**Fig. 5 | TOP-, TOP-deg- and YC-other-initiating transcripts share radiotherapy-responsive dynamics. a**, Illustration of the selection criteria for transcripts identified to initiate with a YR, TOP, TOP-deg or YC-other 5′ initiation makeup. **b**, The YC components of all dual initiators were subdivided into TOP, TOP-deg and YC-other forms as illustrated in **a**, and the number of previously identified DIPs containing >1 TPM of each YC form was calculated; Venn diagram shows the intersection of genes containing >1 TPM of each YC form (generated using Academo software). **c**, The ratio of each YC form to YR transcription in each dual initiator was calculated and divided by the average ratio for that promoter. The frequency distribution of these values, illustrating TOP:YR, TOP-deg:YR and YC-other:YR ratios of transcription for all dual initiators, between responsive (average of CRC1 and CRC2 (gold)) and non-responsive (average of CRC4 and CRC5 (dark blue)) organoid samples is shown. **d**, Analogous to **c**, but comparing the change in dual-initiator TOP, TOP-deg and YC-other component expression upon irradiation between responsive (average of CRC1 and CRC2 (gold)) and non-responsive (average of CRC4 and CRC5 (dark blue)) organoid samples.

(Fig. 3f and Extended Data Fig. 3c), in agreement with clinical samples (Fig. 3b,c). To determine the contribution of YC enrichment and YR depletion within DIPs, we compared the expression of YC transcription and YR transcription separately between organoids (Extended Data Fig. 3d,e). This analysis revealed that the global YC:YR ratio shift between radiotherapy-responsive and non-responsive organoids was predominantly driven by DIPs in which 5′-C transcripts were enriched and 5′-R transcripts were unchanged (1,479 DIPs) in the organoid cohort but with a significant minority of cases in which 5′-R transcripts were depleted and 5′-C transcripts were unchanged (790 DIPs) (Extended Data Fig. 3d,e).

This dynamic radiotherapy response-associated shift in the YC:YR ratio is well demonstrated by the oncogene *SND1* (staphylococcal nuclease and tudor domain containing 1), a gene associated with cancer proliferation, angiogenesis, metastasis and the stress response (reviewed in ref. 41). This dual-initiating promoter displayed a significant change in its TSS usage between responsive, moderately responsive and non-responsive organoids, transitioning from predominantly YC TSS in responsive organoids to balanced transcription in moderately responsive organoids and YR-predominant TSS in non-responsive organoids (Fig. 3g).

As discussed, the investigation of mRNA metabolism previously focused on 79 human ribosomal genes and a small set of translation-associated genes bearing 5′TOP motifs[5]. To investigate the radiotherapy responsiveness of YC-initiating transcripts in these genes, we extracted a definitive gene list from ref. 5, identified genes with sufficient expression (>5 TPM) across all CRC organoid samples and quantified their TSS levels (Supplementary Table 7). This analysis revealed that all of these genes have DIPs, although for the majority,

YC was the predominant initiation class, as expected; the majority of these genes showed radiotherapy responsiveness dynamics in overall gene expression level, but particularly in the 5′-C transcript content with relatively little change in the 5′-R transcript content, in agreement with the data displayed in Fig. 3. Intriguingly, a few showed a reverse dynamic in 5′-C transcript content (YC transcription enriched in the non-responsive cohort), namely *RPL27*, *eiF3F*, *RPS19*, *RPS14*, *RPL7A* and *RPL41*, suggesting that depletion of 5′-C transcript content in non-responsive CRCs was not concurrent with a complete loss of ribosome and translation machinery transcripts but rather a potential switch of TSS (Supplementary Table 7).

The overall dynamics displayed by the majority of known 5′TOP genes, together with the examples in Figs. 1b and 3g and Extended Data Fig. 1a, demonstrated that a large number of DIPs have the capacity to transition their initiation between YC and YR depending on context. This pervasive shift in TSS dynamics represents a hitherto unexplored level of transcript regulation with the potential to impact the post-transcriptional fate of these genes[6–11]. The striking disparity in YC usage in DIPs between responsive and non-responsive CRC tumors and organoids (Fig. 3b–g) suggests that the YC to YR TSS switch could be crucial to our understanding of the differing dynamics between responsive and non-responsive CRC tumors. To identify in which genes these transitions were occurring, we identified 807 DIPs in which YC:YR ratios directly correlated with radiosensitivity (Extended Data Fig. 3f and Supplementary Data 2), with ontology significantly associated with ribosomal, metabolic and biosynthetic processes (Extended Data Fig. 3g,h).

Physiological assessment of the CRC organoids revealed an association between YC enrichment (and radiosensitivity) and faster

proliferation rate (Fig. 3h and Extended Data Fig. 3i), in agreement with the pan-cancer association between YC enrichment and proliferation (Fig. 2c). Furthermore, there was a clear distinction in morphological features among the five organoids. The radiotherapy-resistant lines displayed crypt-like structures within the organoids, reflective of a well-differentiated identity similar to healthy colon organoids (Fig. 3h). The YC-enriched radiotherapy-responsive lines, on the other hand, were cystic in morphology, with no cellular segregation or polarization, previously shown to represent a failure to form crypt-like structures[42,43] (Fig. 3h). This, too, aligns with the pan-cancer analysis shown in Fig. 2, suggesting a link between YC enrichment, cell proliferation and an undifferentiated tumor identity.

### Irradiation depletes YC transcription in responsive organoids

Motivated by the observation of enrichment of initiation of YC transcripts in radiotherapy-responsive CRC tumors, we asked about the potential effect on 5′-C transcript abundance of radiotherapy itself. We challenged the CRC organoid lines with 25 Gy of radiation and performed CAGE-seq. We compared the relative frequency of YC and YR initiation between irradiated and control samples for each organoid (Fig. 4a and Extended Data Fig. 4a). This analysis revealed that YC initiation was depleted relative to YR initiation upon irradiation and that the extent of depletion was again correlated with radiotherapy responsiveness, with a 45%, 28.4% and 4.9% reduction in YC content in responsive (CRC1 and CRC2), moderately responsive (CRC3) and non-responsive (CRC4 and CRC5) organoids, respectively (Fig. 4a and Extended Data Fig. 4a). This global reduction in 5′-C transcripts was again replicated in previously identified DIPs, where YC:YR frequency distribution analysis revealed a depletion in the YC content of DIPs, most marked in radiotherapy-responsive organoids (Fig. 4b).

This YC:YR shift of transcripts was demonstrated well by the dual-initiating promoter of the *C9orf85* gene, linked to cellular differentiation[44] (Fig. 4c). The most abundant transcript isoform in radiotherapy-responsive tumors arose from a YC initiator, in contrast to dominant YR initiation in the moderately responsive and non-responsive cohorts, in line with the dynamic highlighted in Fig. 3. Upon irradiation, however, the 5′-C transcript isoform was specifically depleted in the responsive cohort but unchanged in the moderately responsive cohort and slightly enriched in the non-responsive cohort (Fig. 4c). Besides this example, we surveyed irradiation-associated TSS usage of known 5′TOP-bearing genes. We revisited the 5′TOP gene list (Supplementary Table 7) and compared relative expression between irradiation and control samples for each organoid (Supplementary Table 8). This analysis revealed that the majority of these genes displayed irradiation response-dependent depletion of overall expression but particularly the YC content (Supplementary Table 8).

We then asked which DIPs showed a radiotherapy-responsive transition in TSS usage (YC:YR ratio most depleted upon irradiation in responsive samples to least depleted or enriched in non-responsive samples) (Extended Data Fig. 4b and Supplementary Data 3). This analysis identified 411 genes with gene ontologies associated with

translation, biosynthetic and metabolic processes (Extended Data Fig. 4c,d), similar to the 807 radiosensitivity trajectory genes identified in Extended Data Fig. 3g.

### YC-initiating transcripts share radiotherapy-responsive dynamics

Besides the well-known dependence on the mTOR pathway of 5′TOP mRNAs, a recent study[25] revealed that shorter polypyrimidine stretches could permit mTOR regulation through LARP1-dependant pathways. To investigate whether the radiotherapy-responsive dynamics seen in 5′-C transcripts were due to the differential processing of canonical 5′TOP mRNAs or a more general 5′-C-associated phenomenon, we segregated YC-initiating transcripts into TOP (5 pyrimidines), TOP-deg (4/5 pyrimidines) and YC-other (≤3/5 pyrimidines) transcripts based on their 5′ ends (Fig. 5a). We first surveyed the proportion of DIPs with each of these YC classes (>1 TPM), performing Venn intersection analysis for each 5′-C class (Fig. 5b). This analysis revealed that YC-other transcripts represented the most abundant class, being present (>1 TPM) in 5,747 DIPs (91%), and there was a high degree of overlap between classes, with 4,487 (71%) DIPs containing at least two classes and 2,038 (32%) DIPs containing all three classes. This represents a dramatic increase in the number of genes with 5′TOP-containing transcripts over the <100 5′TOP-containing ribosomal genes[5].

We next asked what part each 5′-C transcript subtype played in radiotherapy response dynamics. We analyzed the change in YC:YR ratio between organoid samples and, upon irradiation, for each YC subclass separately (Fig. 5c,d). All three classes showed the same dynamic separation between the responsive, moderately responsive and non-responsive cohorts. Intriguingly, the extent of separation varied between YC subtypes, with TOP and TOP-deg forms better separating responsive organoids from moderately responsive and non-responsive organoids, while the YC-other subtype was superior for stratifying between the moderately responsive and non-responsive cohorts (Fig. 5c,d).

This finding is noteworthy, as the YC-other transcript format represents a hitherto unexplored transcript type, with no recognized 5′TOP motif imbuing post-transcriptional regulatory properties. However, these YC-other transcripts appeared to show very similar dynamics to those of transcripts containing TOP and/or TOP-deg motifs. One possibility is that the shared dynamics seen in YC-other and TOP or TOP-deg transcripts may be due to downstream 5′TOP motifs in YC-other transcripts (previously identified as TOP-like[15]), permitting post-transcriptional co-regulation with 5′TOP transcripts. Therefore, we investigated the radiotherapy-responsive dynamics of YC-other-initiating transcripts with and without 5′TOP transcripts co-expressed in DIPs (Extended Data Fig. 5a,b) or with and without internal TOP motifs (Extended Data Fig. 5c,d). YC:YR frequency distribution analysis on these two groups identified that the radiotherapy-responsive YC dynamic was retained in DIPs without 5′TOP transcription (Extended Data Fig. 5a,b) and regardless of the presence of an internal TOP motif (Extended Data Fig. 5c). YR transcripts with internal TOP sequences versus those without such sequences

**Fig. 6 | A YC-defined gene signature marks radiotherapy response. a**, Venn diagram of the pan-cancer trajectory genes (described in Extended Data Fig. 1d), radiosensitivity trajectory genes (described in Extended Data Fig. 3f) and irradiation-affected trajectory genes (described in Extended Data Fig. 4b). The 147 genes forming the intersection of the latter two groups are termed the radiotherapy responsiveness signature genes (Venn diagram generated with Meta-Chart software). **b**, Intersection of MSigDB pathway enrichment ontology analysis of the three trajectory gene sets detailed in **a**. Dotted line shows 0.05 FDR threshold. **c**, UCSC Genome Browser view of an irradiation-responsive dual promoter, *GMPR2* (guanosine monophosphate reductase 2), highlighting the location of the annealing sites for primers designed to segregate the expression of the YC and YR components through RT−qPCR analysis. Direction of transcription (on the reverse strand) is illustrated by arrows. **d**, Bar graphs

showing the relative expression of the YC and YR components of five candidate dual-initiation genes in the CRC organoids and upon irradiation, calculated by RT−qPCR analysis. ($n = 2$ responsive, $n = 1$ moderately responsive and $n = 2$ non-responsive organoids; *$P < 0.05$, two-tailed unpaired $t$-test, statistically analyzed comparisons were responsive versus non-responsive and control versus irradiated; data are presented as mean values ± s.e.m., full list of $P$ values available in the Source data). **e**, Bar graphs showing the relative expression of the YC and YR components of five candidate dual-initiation genes in 12 CRC clinical samples (six responsive and six non-responsive independent biological samples), calculated by RT−qPCR analysis. ($P < 0.0001$, 0.0001, 0.0097 and 0.0046 for *C9orf85*, *PRORP*, *SND1* and *GMPR2*, respectively; two-tailed unpaired $t$-test, statistically analyzed comparisons were responsive versus non-responsive and control versus irradiated; data are presented as mean values ± s.e.m.).

showed no irradiation-responsive dynamic (Extended Data Fig. 5d). Thus, transcript radiotherapy response was primarily dependent on YC initiation and less on the presence of a classic or internal TOP motif, similar to suggestions of mTOR-dependent post-transcriptional RNA regulation mediated by pyrimidine stretches shorter than the canonical TOP motif[25,45]. This suggests that segregating these YC subtypes for sensitive stratification of tumors may be beneficial. However, as the

transcript dynamics described throughout this investigation were shared by all YC classes, we continued to focus on YC dinucleotide dynamics for subsequent analyses.

**A YC-defined gene signature marks radiotherapy response**
We next asked whether there is a union of genes among which the 5′-C transcript content is responsive to cancer features (differentiation

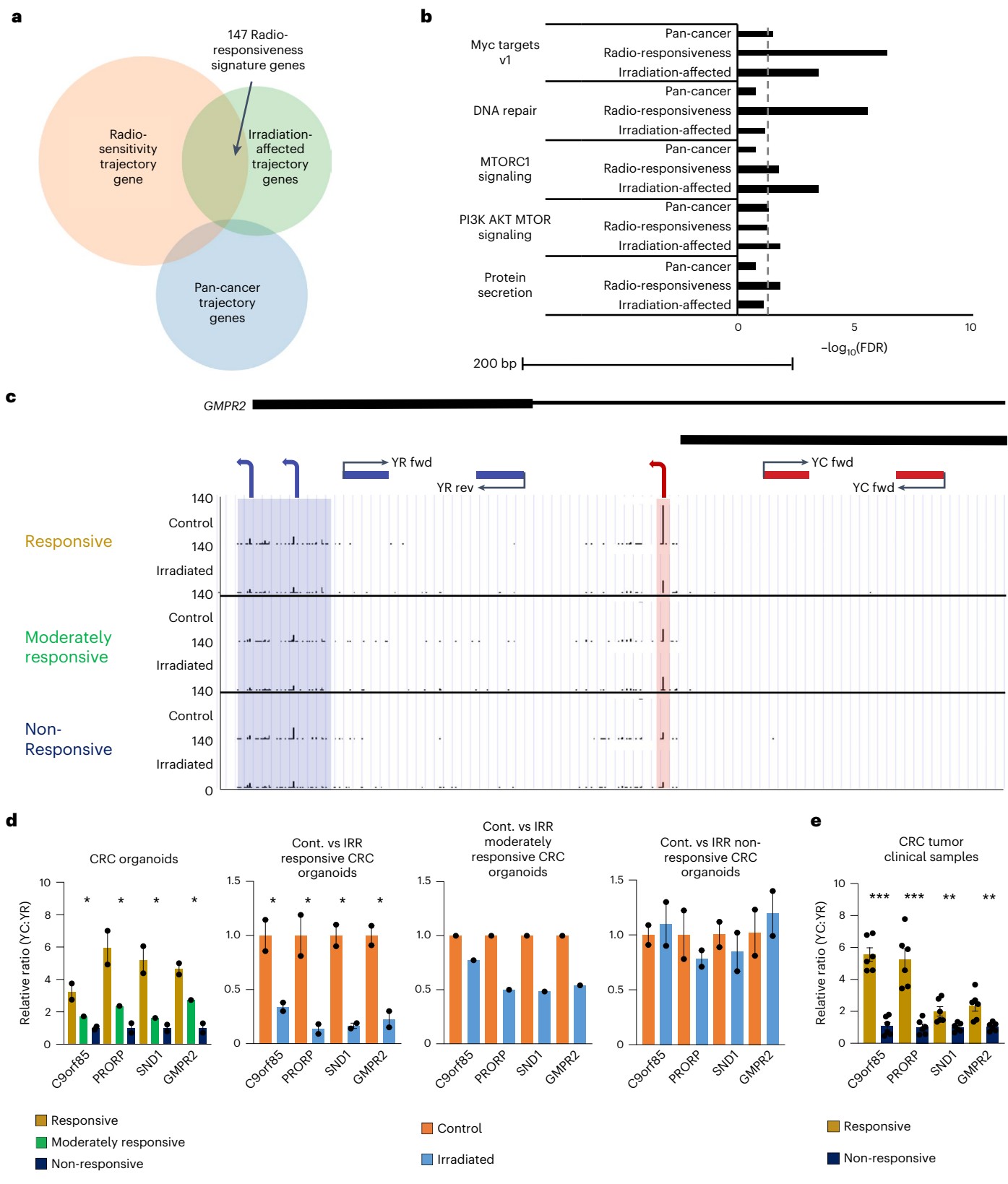

status, CRC radiotherapy responsiveness, irradiation response). We performed Venn overlap analysis of the pan-cancer, radiosensitivity and irradiation-affected trajectory gene sets (Extended Data Figs. 1d, 3f and 4b), identifying a modest overlap (ten genes) between all (Fig. 6a and Supplementary Data 5), suggesting that the types of genes displaying differential 5′-C transcript regulation are highly context-dependent. However, further ontology analysis intersecting these gene sets with the Molecular Signature Database (MSigDB) identified overlap in associated regulatory pathways (Fig. 6b and Extended Data Figs. 1e, 3h and 4d). Significant or near-significant enrichment for genes regulated by Myc, DNA repair, mTORC1 signaling, PI3K−Akt−mTOR signaling and protein secretion in all three gene sets was found (Fig. 6b), identifying these pathways as potential regulators of 5′-C transcript metabolism. Investigation of genes shared between the YC-modulated gene sets identified a striking overlap in Myc regulatory targets (six of ten genes; Supplementary Table 9).

In contrast to the small number of genes shared by the pan-cancer, radiosensitive and irradiation-affected trajectory gene sets, there was a significant overlap (147 genes) between the latter two gene sets (Fig. 6a). This overlap would represent a putative gene signature in which transcription initiation dynamics are directly related to radiotherapy response in CRC. Gene ontology again revealed a strong association with translational and biosynthetic processes and significant involvement of Myc and PI3K−Akt−mTOR regulatory pathways as well as a borderline-significant association with the p53 pathway (Extended Data Fig. 6a,b and Supplementary Data 4), in agreement with Extended Data Figs. 3g,h and 4c,d. Further analysis of the gene ontologies within this gene set identified a broad range of functions represented, including metabolism, gene expression and cell proliferation (Extended Data Fig. 6c). We termed these the 'radiotherapy responsiveness signature' gene set, representing an ideal group in which to investigate the regulatory background of 5′-C transcript modulation. We sought clues for potentially shared transcriptional regulation by transcription factor binding site enrichment analysis. We found enrichment for ETS like-1 protein (Elk1) and GA-binding protein alpha (Gabpa) transcription factor binding sites within their promoter regions (Extended Data Fig. 6a). These two ETS-related transcription factors have both been identified as effectors in Myc and mTOR regulatory pathways, including in CRC[46–48]. To explore this further, we investigated the expression profiles of these transcription factors between the organoid samples, alongside Myc, p53 and Tbpl1 (also called Trf2), the only transcription factor to have previously been implicated in the regulation of YC transcription[24]. This analysis revealed that each of these transcription factors showed a similar expression pattern and trajectory among the CRC organoid samples to the radiotherapy response signature gene set (Extended Data Fig. 7a). Irradiation of the organoids did not significantly affect the expression of these transcription factors with the exception of p53, whose expression was enhanced, but to the greatest extent in the non-responsive cohort (Extended Data Fig. 7b). The promoters of these transcription factors each have dual initiation, with the YC component varying from 4.3–67.6% of total transcription and showing direct correlation with radiotherapy response (Extended

Data Fig. 7a). This was exemplified by the *TBPL1* promoter, whereby the predominant transcript class switched from YC in the responsive cohort to YR in the non-responsive cohort, with balanced transcription in the moderately responsive cohort (Extended Data Fig. 7c). To investigate whether Myc and p53 could be having a direct regulatory role in transcription of the radiotherapy response signature gene set and to validate the gene ontology identification of Elk1 and Gabpa as candidate regulators, we performed motif enrichment analysis for each of these factors, comparing the prevalence of their JASPAR consensus motifs between the promoters of the radiotherapy response signature gene set versus the background of all identified promoters in our analysis (Extended Data Fig. 7d). This revealed a significant enrichment for the Elk1, Gabpa and Myc binding motifs within the promoters of the radiotherapy response signature gene set ($P < 0.002$–0.049) (Extended Data Fig. 7d,e).

To investigate whether this radiotherapy responsiveness signature could be used to segregate responsive and non-responsive tumors without the need for CAGE-seq, we generated a quantitative PCR with reverse transcription (RT−qPCR) protocol that separately monitors YC and YR transcription from DIPs. Four candidate genes (*C9orf85*, protein only RNase P catalytic subunit (*PRORP*), guanosine monophosphate reductase 2 (*GMPR2*) and *SND1*) from the radiotherapy responsiveness signature gene set were selected, in which YC and YR initiation are sufficiently spatially separated to allow discriminating primer design. We designed primers to amplify the longest transcripts, which in each case were initiating from 5′-R. These 5′-R transcript levels were subtracted from the total RNA detected by internal primers to indirectly calculate the YC contribution to the total (Fig. 6c, Extended Data Fig. 8 and Supplementary Table 10). We used this RT−qPCR approach to compare the YC:YR ratio in selected DIPs among the five CRC organoid lines (Fig. 6d and Supplementary Table 5). This showed significant enrichment of 5′-C transcripts in responsive versus non-responsive organoid samples (Fig. 6d), in agreement with CAGE-seq analysis. Again, similarly to CAGE-seq, 5′-C transcripts were significantly depleted upon irradiation in responsive organoids, unlike in moderately responsive and non-responsive organoids (Fig. 6d). We next investigated the prognostic potential of this RT−qPCR approach to distinguish CRC clinical tumor samples based on their chemo-radiotherapy responsiveness, testing six responsive and six non-responsive clinical tumor samples, collected as described in Extended Data Fig. 3a and Supplementary Table 5. This analysis revealed a significant enrichment in the YC:YR ratio in the responsive tumor samples over the non-responsive samples for each gene, but particularly for *C9orf85* and *PRORP* (Fig. 6e).

## CRC radiosensitization enhances 5′-C transcript abundance

The therapeutic inhibition of the PI3K−Akt−mTOR pathway has previously been shown to radiosensitize a range of cancers, including CRC[39,49–51]. This highlights the possibility that modulation of 5′-C transcript abundance by PI3K−Akt−mTOR may have a role in this radiosensitization. To test this premise, we treated CRC organoids from the radiotherapy-resistant cohort (CRC4 and CRC5) with the PI3K−Akt−mTOR pathway inhibitor dactolisib with and without radiotherapy.

**Fig. 7 | Inhibition of PI3K−AKT−mTOR pathway signaling enhances YC transcript abundance and restores radiotherapy-induced transcriptional dynamics. a**, Bar graph of survival of organoids from each resistant line, exposed to each experimental condition, relative to untreated organoids ($n = 3$ independent experiments, *$P < 0.05$, **$P < 0.01$, ***$P < 0.001$, ****$P < 0.0001$; ordinary one-way ANOVA with Tukey's multiple comparisons test; data are presented as mean values ± s.e.m., full list of $P$ values available in the Source data). **b**, Frequency distribution of YC:YR ratios of all dual initiators ($n = 6,292$) and radiotherapy-responsive signature genes ($n = 147$) in CRC5 control (orange) and dactolisib-treated (brown) samples relative to the average ratio between samples. **c**, Frequency distribution of the fold change in YC:YR ratios upon irradiation of the radiotherapy-responsive signature genes in CRC5 control

(orange) and dactolisib-treated (brown) samples as well as responsive (average CRC1 and CRC2 (dotted gold)) and moderately responsive (CRC3 (dotted green)). **d**, UCSC Genome Browser view of *C9orf85*. This promoter shows an enrichment of the YC component upon radiosensitizing PI3K−mTOR inhibition (dactolisib) treatment in non-responsive tumors (CRC5) and a restoration of YC depletion upon irradiation and dactolisib treatment compared to dactolisib alone. **e**, Bar graphs of RT−qPCR analysis of relative YC and YR expression from candidate dual initiators in CRC organoids treated with dactolisib, irradiation or a combination ($n = 3$ independent experiments, *$P < 0.05$, **$P < 0.01$, ***$P < 0.001$, ****$P < 0.0001$; ordinary one-way ANOVA with Tukey's multiple comparisons test; data are presented as mean values ± s.e.m.; full list of $P$ values available in the Source data).

We used 0.1 µM dactolisib, the previously identified IC$_{50}$ dose when combined with radiotherapy, but with minimal survival impact alone[39] (Fig. 7a).

CAGE-seq analysis was performed on CRC5 samples treated with dactolisib (± irradiation) together with untreated (control) and irradiation-treated samples (Fig. 7b and Extended Data Fig. 2a,b). Analysis of the effect of dactolisib on the YC:YR content of all DIPs revealed a

modest effect of drug treatment (Fig. 7b). However, when assaying the radiotherapy responsiveness signature gene set, the effect of dactolisib treatment (enrichment of the YC content of DIPs) was significantly greater (Fig. 7b), suggesting a role for PI3K–Akt–mTOR pathway signaling in the regulation of this radiotherapy-responsive cohort of DIPs. Analysis of the effect of dactolisib treatment on the irradiation response of this signature revealed that PI3K–Akt–mTOR therapeutic blockade

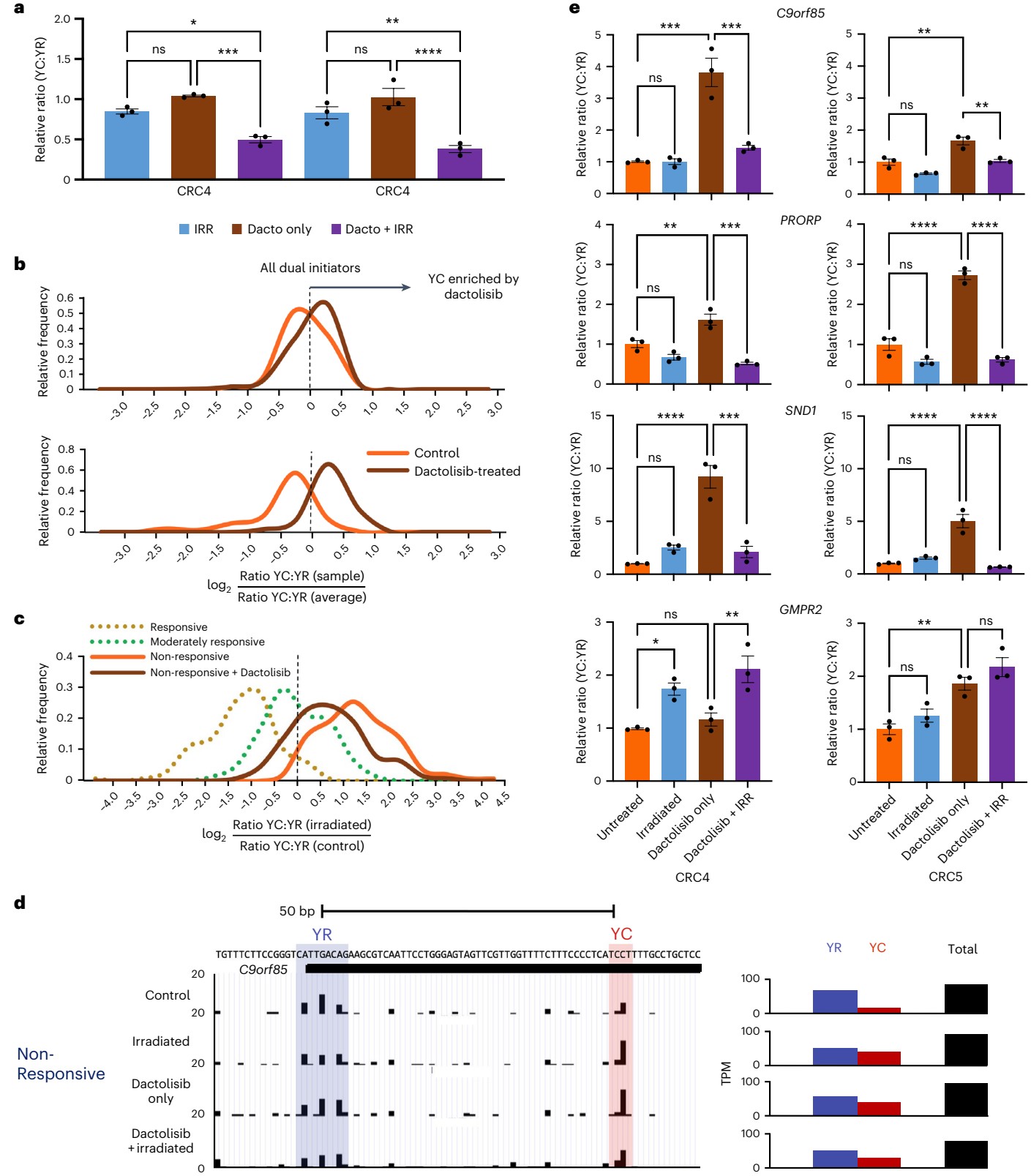

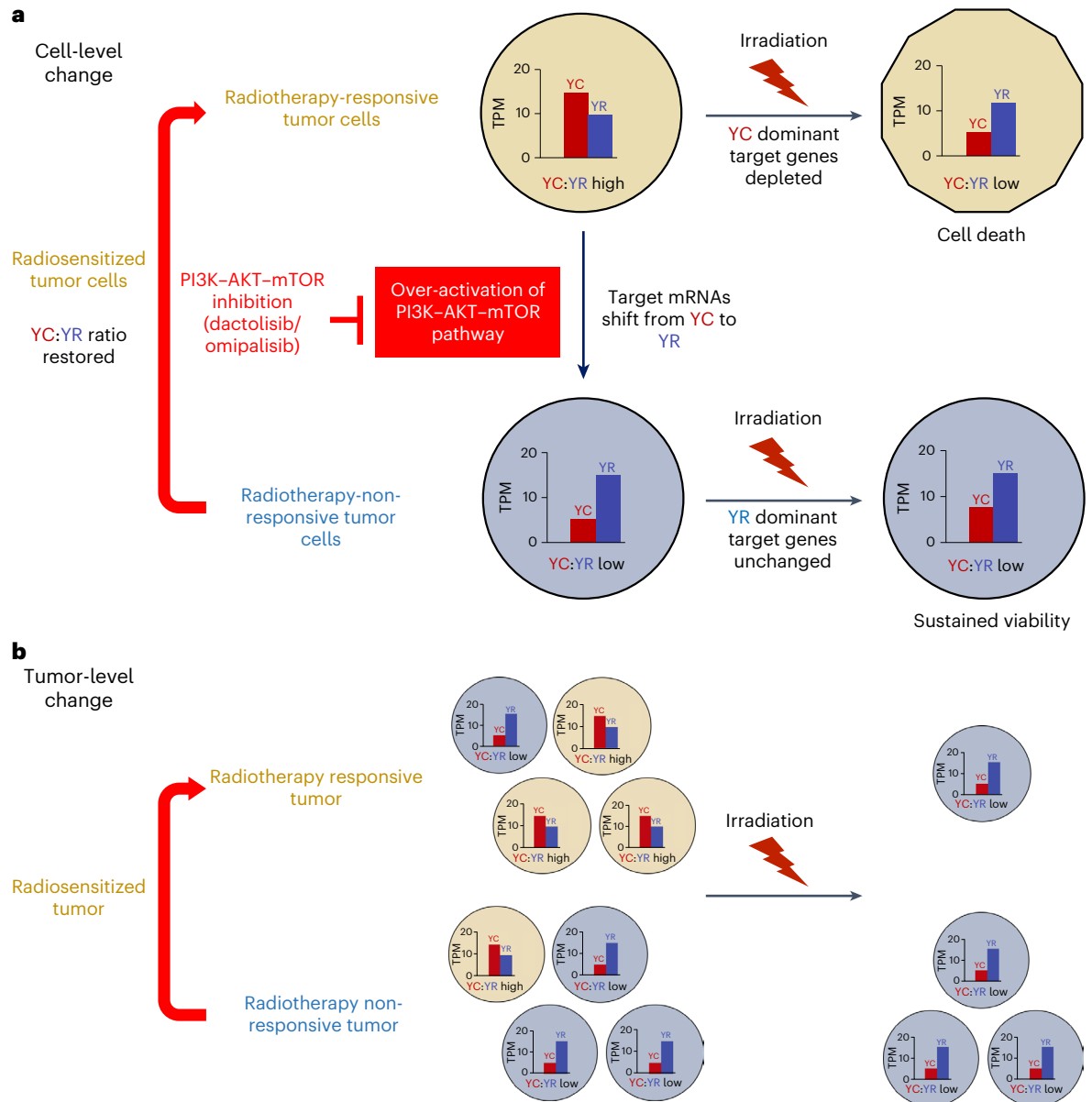

**Fig. 8 | Model diagram of YC:YR dynamics in irradiated CRC organoids. a,** Aberrant PI3K–AKT–mTOR and Myc pathway activity may shift the transcript balance of dual-initiator genes towards radiotherapy-resistant YR predominance, curtailing radiotherapy sensitivity in non-responsive tumor cells. This is reversed and the YC:YR balance is restored by PI3K–AKT–mTOR inhibition using dactolisib or omipalisib, radiosensitizing the tumors. **b,** At the tumor level, radiotherapy-sensitive tumors are predominantly made up of YC-enriched, radio-responsive cells and radiotherapy-non-responsive tumors are predominantly made up of YC-depleted, radiotherapy-resistant cells. Upon irradiation, YC-enriched cells are selectively killed, leaving only radiotherapy-resistant YC-depleted cells. This leads to a significant shift in YC transcript levels in radiotherapy-responsive tumors and little change in radiotherapy-resistant tumors. PI3K–AKT–mTOR inhibition drives a cell-level enrichment of 5′-C transcripts, shifting the balance between YC-enriched and YC-depleted cells in the non-responsive tumor, rendering it more radiosensitive.

enhanced the YC content depletion upon irradiation in non-responsive organoids, bringing it in line with that seen in moderately responsive organoids (Fig. 7c). The dynamics of the dactolisib treatment-induced YC modulation were also visible at the individual gene level (Fig. 7d). To further validate these findings, we performed RT–qPCR YC:YR ratio quantification for each of the treatment groups. This analysis confirmed that dactolisib enhanced 5′-C transcripts and restored the irradiation-responsive dynamics (Fig. 7e), in agreement with Fig. 7b,c.

Dactolisib is also a potent inhibitor of DNA-PK, ATM and ATR, master regulators of the DNA damage response, although at a 2-fold-, 3.5-fold- and 10-fold-lower specificity, respectively, than mTOR and PI3K[52]. Additionally, the radiotherapy responsiveness, irradiation-affected and pan-cancer trajectory gene sets were all

significantly enriched for DNA damage-responsive genes (Fig. 6b). Thus, DNA damage response may contribute to the drug-induced radiosensitization and may also be involved in the YC:YR ratio modulation effect. To dissect this possibility, we repeated the radiosensitization experiment described in Fig. 6 using the PI3K–mTOR dual-inhibitor omipalisib and the ATM/ATR inhibitor VE821. It should be noted that due to the considerable crosstalk between the PI3K–Akt–mTOR and DNA damage response pathways, neither of these drugs are entirely specific to their respective pathways. Omipalisib additionally inhibits DNA-PK (with tenfold-lower specificity than PI3K) and VE821 downregulates mTOR at high concentrations (10 µM), as a potential off-target effect[53]. The drugs were used at previously published radiosensitizing doses of 0.1 µM and 1 µM for omipalisib and VE821, respectively[54,55],

and both induced significant radiosensitization in the CRC4 and CRC5 radiotherapy-resistant lines, similar to dactolisib (Extended Data Fig. 9a,b). Strikingly, both drugs induced the same modulation of YC:YR ratios to dactolisib (Extended Data Fig. 9c). In summary, therapeutic blockade of both PI3K–Akt–mTOR and DNA damage pathway signaling shifted the 5′-C transcript levels of signature genes in radiotherapy-resistant CRC lines towards that in more radiosensitive lines. Upon irradiation, this YC component was significantly depleted, displaying a dynamic similar to that seen in irradiated radiosensitive CRC lines, proportional to the survival impact of the irradiation treatment, highlighting the utility of this gene set as a sensitive predictor and/or reporter of irradiation response in CRC organoids (Fig. 8).

## Discussion

This study demonstrated a previously unexplored layer of gene regulation in cancers, identifying TSS selection as a regulatory interface between transcription and translation-associated post-transcriptional RNA fate. We used CAGE-seq to profile TSS selection in thousands of genes with a range of functions, producing transcripts from the same promoter but with differing 5′ ends[23], a transcript variation nuance invisible to RNA-seq. We focused on the two most abundant transcript 5′ isoforms (initiating from pyrimidine:purine (YR) and pyrimidine:cytosine (YC) dinucleotide motifs), showing these to be differentially regulated between matched cancer and healthy tissues but also between cancers of differing differentiation status and radiotherapy sensitivity, allowing highly sensitive segregation of cancers into poorly, moderately and well-differentiated states and CRCs into highly radiotherapy-responsive, moderately responsive and non-responsive cohorts. This reveals that both of these cellular events are marked by global changes in RNA transcript metabolism, mediated by TSS shift.

Our results demonstrate a distinct molecular signature of radiotherapy responsiveness, characterized not by overall gene expression but rather by the contribution to the transcriptome of TSS isoforms. We showed YC initiation to be severely depleted in response to irradiation, in contrast to the moderately changing YR-initiated transcript level in CRC organoids. This YC class is additionally enriched in radiotherapy-responsive clinical CRC samples and organoids derived from them. Strikingly, when we investigated gene sets with the most dynamic YC initiation components in response to radiation, they marked the same genes enriched in radiotherapy-responsive tumors, suggesting a mechanistic role in radiotherapy response.

We demonstrated that a molecular signature of radiotherapy responsiveness can be measured by custom RT–qPCR and that this approach dissected responsive and non-responsive tumors to chemo-radiotherapy. Furthermore, we showed that radiosensitization of CRC tumors by PI3K–Akt–mTOR and DNA damage pathway blockade resulted in the enrichment of YC-initiating transcripts in radiotherapy-resistant tumors, closer to the level of radiosensitive lines, establishing a mechanistic link between CRC radiosensitivity, the PI3K–Akt–mTOR pathway, the DNA damage pathway and YC transcription initiation. As healthy tissues are generally relatively YC depleted, this approach may also radiosensitize them, with the potential for off-target toxicity needing further investigation. As mTOR inhibition selectively blocks the translation of 5′TOP-containing transcripts[15], this could be due to stabilization of mRNAs. DNA damage also has been shown to repress the RNA translation and ribosomal genes in yeast[56–58] and in humans through p53 signaling[59,60]. Collectively, our results demonstrate the prognostic value of this novel molecular signature that can be cost-effectively monitored to indicate CRC tumor response to radiotherapy and the efficacy of radiosensitizing therapies.

Our results identified a much larger population of genes that produce 5′TOP mRNAs with potential for mTOR signaling regulation than previously described[5]. Since only TSS-resolving technologies can detect the specific effects on 5′-C transcript dynamics, it is feasible that mTOR-mediated targeting of the YC component of DIPs was missed by conventional transcriptomics. Thus, the YC component of thousands of DIPs may be targeted by mTOR signaling-associated translation regulation[12–17]. It is noteworthy that the molecular function of the genes with differential 5′-C transcript metabolism extends beyond that of the canonical 5′TOP mRNAs. We show roles as diverse as DNA replication, transcriptional regulation and mitochondrial function, with YC:YR transcripts under distinct regulatory dynamics dependent on cellular context. This highlights the importance of a previously little-appreciated regulation by TSS selection, which permits the co-regulation of a broad range of transcripts, mediated by their shared 5′ end. It also demonstrates the power of CAGE-seq (and other 5′ end-resolving techniques) to decipher transcript identities and intra-promoter transcript dynamics with far greater specificity than RNA-seq.

In this study, we highlight the putative role of the interconnected PI3K–Akt–mTOR, Myc, p53 and DNA damage signaling pathways in radiosensitivity-associated differential YC:YR metabolism. The PI3K–Akt–mTOR signaling pathway modulates 5′TOP mRNA interactions with effector proteins and enhances their translation[6–11,45,61]. Loss of the tumor suppressor p53 was found to enhance 5′TOP mRNA translation through mTORC1 signaling[62]. DNA damage-responsive Myc signaling was also associated with the regulation of transcript metabolism. Myc inactivation was associated with decreased mRNA translation, in particular, mitochondrial respiration genes[63], which were represented in the molecular signature of radiotherapy responsiveness[63]. Additionally, Myc regulates ribosome biogenesis through the recruitment of RNA polymerase II to ribosomal protein genes to produce 5′TOP mRNAs[18,64]. This suggests that Myc may not only regulate the transcription of 5′TOP but also regulate other YC-initiating mRNAs emanating from DIPs. Indeed, we showed that genes of the radiotherapy responsiveness molecular signature are significantly enriched for Myc binding sites, alongside transcriptional effectors associated with Myc and PI3K–Akt–mTOR signaling cascades[46–48]. Elk1 has been shown to directly induce *MYC* gene expression in CRC tumorigenesis[46], while target genes of Gabpa are specifically enriched for metabolic, stress response, DNA damage and MYC-regulated oncogenic signatures[47]. Tbpl1, a general transcription factor previously implicated in the transcriptional regulation of 5′TOP-bearing ribosomal genes[24], was also found to be co-regulated with radiotherapy-associated YC transcriptional dynamics. The enriched expression of Tbpl1 in radiotherapy-responsive organoids is in line with enriched 5′-C transcript abundance and suggests a potential role for Tbpl1 in the regulation of radiotherapy-responsive 5′-C transcript metabolism.

A striking finding from the investigation of candidate regulators of YC transcription (Myc, Gabpa and Elk1) was that their genes carry both YC and YR initiation products, with the expression of their 5′-C transcripts correlated with radiosensitivity and that of the 5′-C transcripts of the molecular signature genes. This suggests a possible regulatory feedback loop mediated by TSS selection and a potential role for both transcriptional and post-transcriptional regulation through PI3K–Akt–mTOR signaling in YC–YR dynamics. It is likely that both layers of regulation play a role in radiotherapy response-associated 5′-C transcript metabolic changes. Further investigation to unpick the role of transcription versus post-transcriptional mRNA stability in these dynamics will be necessary.

Taking these findings together, we propose YC–YR initiation dynamics as a read-out of transcriptional and post-transcriptional mechanisms through which a range of proliferation-associated signaling pathways, including PI3K–Akt–mTOR–Myc, regulate tumor cells. Aberrant PI3K–Akt–mTOR–Myc pathway signaling in radiotherapy-resistant CRCs changes the metabolic landscape of the cell and shifts the balance of transcript abundance from YC to YR initiation. As YR-transcripts are less affected by ionizing radiation-responsive metabolic changes in transcript processing and stability, this represents a reduction in the radiosensitivity of the transcriptome, constituting a survival benefit (Fig. 8). Upon irradiation,

**Article**

YC-enriched radiosensitive cells are selectively killed, reducing the YC content of the post-treatment culture, particularly in radiosensitive organoid lines where this population predominates, but with minimal effect on radiotherapy-resistant YC-depleted cells, predominant in the radiotherapy-resistant cultures. Upon PI3K–Akt–mTOR blockade, 5′-C transcript metabolism in the resistant cells partially transitions to a more radiosensitive program, rendering more cells in the culture radiosensitive and, thus, the organoids become radiosensitized (Fig. 8).

## Online content

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

## Methods

### Ethics approval

This project received approval (code 17-287) from the Human Biomaterials Resource Centre, Birmingham, United Kingdom, under ethical approval from the Northwest–Haydock Research Ethics Committee (reference 15/NW/0079).

### Patient-derived tissue and organoid samples

FFPE pre-treatment biopsy specimens from patients with primary colonic or rectal adenocarcinoma were collected with the help of the University of Birmingham Human Biomaterials Resource Centre (HBRC). Anonymized clinicopathological data such as patient demographics, neoadjuvant treatment status, tumor location, TNM stage and histopathological data including tumor regression grade were obtained, again with the help of HBRC. Organoid samples used in this study were generated as described in ref. [39]. Ethical approval for all tissue collection was encompassed under HBRC application 17-287 (ethical approval reference 15/NW/0079).

### Organoid maintenance

Organoids were split once per week and disassociated to a one-cell suspension using TrypLE before being washed in PBS and re-plated in 24-well plates at a concentration of 50,000 cells per well in 50 µl of 8 mg ml$^{-1}$ basement membrane extract type 2 (Bio-techne). Organoids were maintained in 500 µl of IntestiCult medium (STEMCELL Technologies) containing Primocin (Invitrogen) at a final concentration of 100 µg ml$^{-1}$, which was replenished twice per week, and were kept in 5% $CO_2$ at 37 °C.

### Organoid doubling time assessment

A total of 50,000 cells were plated per line in 24-well plates on day 0, disassociated on day 4 or day 9 and counted again using a Bio-Rad TC20 automated cell counter three times per well. Doubling time was calculated using the following formula:

$$\text{Doubling time} = [T \times (\ln 2)]/[\ln(X/50,000)]$$

where $R$ = time/days (4 or 9) and $X$ was the average count on day 4 or 9.

### Organoid radiotherapy and radiosensitization assessment

A total of 50,000 cells were plated per line in 24-well plates on day 0. When radiosensitizing drug treatment with dactolisib, omipalisib or VE821 (Selleck Chemicals) was performed, this was added with a medium change on day 3. Five sequential days of 5 Gy of irradiation commenced on day 4, terminating on day 9. Drugs were replenished with a medium change on day 6 and removed with a medium change on day 9. This was followed by a 5-day recovery period. Cell viability was assessed in triplicate using the CellTiter-Glo 3D Cell Viability Assay (Promega) following the manufacturer's protocol.

### Organoid imaging

For morphological assessment of organoids, each line was grown for 14 days in normal culturing conditions and fixed following a high-resolution fixing protocol[65]. On day 14, organoids were recovered from basement membrane extract with 500 µl of cell recovery solution (Corning) on ice, washed with PBS–BSA and fixed using 4% PFA. Samples were then loaded into low-melting point agarose (Invitrogen) and imaged with a Zeiss Z1 lightsheet microscope.

### Extraction of RNA from tissue and organoid samples

RNA was extracted from FFPE tissue samples using the RNeasy FFPE Kit (Qiagen) following the manufacturer's protocols. RNA was extracted from fresh harvested organoid samples using the miRNeasy mini Kit (Qiagen) following the manufacturer's protocols, and RNA quality was analyzed by capillary electrophoresis (Bioanalyzer 2100, Agilent). All samples had an RNA integrity number of >9.

### CAGE library preparation and sequencing

CAGE libraries were generated from FFPE RNA samples following a previously published protocol[38]. Libraries were individually barcoded and pooled for sequencing on the Illumina NextSeq, using the High 75 cycle single-read run operation program. CAGE libraries were generated from CRC organoid RNA samples following a previously published protocol[66]. To increase the multiplexing capacity and demultiplexing efficiency of libraries sequenced on Illumina two-color instruments (for example, NextSeq or NovaSeq), the sequencing adaptors listed in the above protocol were modified to use the standard Illumina barcoding and indexing strategy (instead of the legacy 3-bp barcode at the beginning of read 1), in this case, 8-bp TruSeq Unique Dual Indexes. To attach the Illumina P5/P7 flow cell adaptors and I5/I7 indexes, an additional amplification step of five to six PCR cycles was performed using NEBNext Ultra II Q5 Master Mix (New England BioLabs). The PCR reactions were purified with 1.4× AMPure XP beads (Beckman Coulter) to yield the final libraries. Libraries were sequenced (2 × 50-bp reads) on a NovaSeq 6000 v1.5 SP 100-cycle flow cell.

### Publicly available CAGE-seq data

Mapped human CAGE-seq and RNA-seq data were downloaded from the FANTOM5 consortium database[29,30].

### CAGE mapping and CTSS calling

The human genome assembly (GRCh38/hg38) was downloaded from the UCSC Genome Browser[67]. For all newly generated CAGE samples, reads were trimmed to remove the linker and unique molecular identifier regions (if applicable). Reads were mapped using Bowtie (v.1.3.1)[68], allowing a maximum of two mismatches and only uniquely mapping tags with a MAPQ value of >20. The R/Bioconductor package CAGEr (v.1.34.0) was used to remove the additional G nucleotide, due to the CAGE protocol, where it did not map to the genome[31]. All unique 5' ends of reads were defined as a CTSS and reads were counted at each CTSS per sample. These raw read counts were subsequently normalized based on a power-law distribution based on 10$^6$ reads[69] and defined as normalized TPM.

### Calling transcriptional clusters

CTSS that were supported by at least 0.5 TPM in one of the samples were clustered based on a maximum allowed distance of 20 bp between two neighboring CTSS. These transcriptional clusters were then trimmed on the edges to obtain more robust boundaries of transcriptional clusters by obtaining the positions of the 10th and 90th percentiles of expression per transcriptional cluster. Only transcriptional clusters with higher than 5 TPM expression were considered. Finally, transcriptional clusters across all samples were aggregated if they were within 100 bp of each other to form consensus clusters for downstream analyses.

### Annotation

The consensus clusters were annotated to genomic features using the CAGEr function 'annotateCTSS' in conjunction with the R/Bioconductor package rtracklayer (v.1.52.1)[70]. Consensus clusters mapping to the promoter region of Ensembl (hg38)-annotated genes were selected for further analysis. Where multiple consensus clusters mapped to the same gene, their TPM expression values were merged.

### Calling YR, YC, 5'TOP and internal TOP transcripts

CTSS within consensus clusters were segregated on the basis of their +1 or −1 base configuration into YR (CG, CA, TG, TA) and YC (CC, TC) classes. For further segregation of CTSS, the first five bases of the transcript were used. Transcripts with a CYYYY configuration were classed as '5'TOP', transcripts with a C(3Y1R) configuration were classed as '5'TOP-deg' and all other YC-initiating transcripts were classed as 'YC-other'. For the assessment of internal TOP transcripts, the first

50-bp region of each transcript was assessed by sliding window analysis for an unbroken stretch of five pyrimidines. Transcripts initiating from a YC (excluding 5′TOP) or YR dinucleotide and with an unbroken stretch of five pyrimidines in the first 50 bp were identified as YC internal TOP and YR internal TOP transcripts, respectively. Transcripts with a 5′TOP identity were not assessed for the presence of an internal TOP.

### Differential gene expression analysis

The raw read counts were extracted for the consensus clusters and collapsed into total count per consensus cluster. The DESeq2 (v.1.32.0) R/Bioconductor package[71] was used to define differential expression, and the threshold of differential expression was set at an adjusted $P$ value of <0.05. These results were cross-referenced to the consensus cluster information of samples. In cases of more than one consensus cluster mapping to the region, the consensus cluster with the highest expression was chosen to represent the region.

### RT–qPCR validation of candidate YR–YC dual promoters

cDNA was generated from sample RNA using the SuperScript VILO cDNA Synthesis Kit (Invitrogen), following the manufacturer's conditions. RT–qPCR was then performed using the PowerUp SYBR Green system (Applied Biosystems), following the manufacturer's conditions, and was run on the Real-Time PCR ABI 7900HT machine (Applied Biosystems). RT–qPCR primer sequences are provided in Supplementary Table 10.

### Gene ontology analysis

Gene ontology analysis was performed using the ShinyGO 0.77 online program (http://bioinformatics.sdstate.edu/go, accessed 26 July 2023) and Panther classification system for Extended Data Fig. 6b (http://pantherdb.org/webservices/go/overrep.jsp, accessed 28 September 2022).

### Core promoter motif enrichment analysis

Position weight matrices for Elk1 (MA0028.2), Gabpa (MA0062.1), Myc (MA0147.3) and p53 (MA0106.1) binding site motifs were obtained from converting frequency matrices from JASPAR (9th release; 2022; ref. 72). Each consensus cluster was centered on the most expressed CTSS (the dominant TSS), and each sequence was scanned from 150 bp upstream and 50 bp downstream. A hit was reported if the scanned region contained a sequence with a 90% match to the position weight matrix. Occurrence was counted for the radiotherapy response signature genes and compared to all other consensus clusters. Significance was assessed using Fisher's exact test. Obtained $P$ values were considered significant at <0.05.

### Pan-cancer mutation analysis

The mutation status for each of the cancer cell line samples listed in Supplementary Table 1 was extracted from the Cancer Cell Line Encyclopedia (Broad Institute)[32]. Frequency of mutation in each of the cancer cohorts was assessed and chi-squared analysis was performed to identify mutations with significant association with either the YC-enriched or YC-depleted cancer cohorts.

### Statistics and reproducibility

Statistical analyses were performed on GraphPad Prism (v. 9) unless otherwise stated in the legend. No statistical method was used to predetermine sample size, as this was instead determined by maximum sample availability, which was limited in the case of organoids by numbers successfully generated clinical samples, by ethics agreements and previously published FANTOM5 CAGE data by cancer samples having a suitable match to healthy tissue samples. No data were excluded from the analyses. The experiments were not randomized and the investigators were not blinded to allocation during experiments and outcome assessment, as all analyses were objective in nature. The only subjective analysis was the organoid morphology scoring in

Fig. 3h; however, with a sample size of five, blinding was not practicable, and images of each organoid are shown for the reader to make their own judgment.

### Figure generation

All figures were generated on GraphPad Prism, Microsoft Excel or Microsoft PowerPoint unless otherwise stated in the legend. Venn diagrams were generated using Academo software (https://academo.org/demos/venn-diagram-generator) for Fig. 5b and Meta-Chart software (https://www.meta-chart.com/venn#/display) for Fig. 6a (both last accessed 5 October 2023).

### Reporting summary

Further information on research design is available in the Nature Portfolio Reporting Summary linked to this article.

## Data availability

All relevant data and results included in this article have been published along with the article and its Supplementary Information and Source data files. Raw sequencing data for CAGE-seq is publicly available at NCBI Sequence Read Archive under accession number PRJNA934878. Other relevant data can be obtained, upon reasonable request, from the corresponding authors. Source data are provided with this paper.

## Code availability

CAGE data were analyzed using Bioconductor package CAGEr following established pipelines, available from https://bioconductor.org/packages/release/bioc/vignettes/CAGEr/inst/doc/CAGEexp.html. Custom code used for this analysis is provided in the Supplementary_code_R document.

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

## Acknowledgements

We thank The Human Biomaterials Resource Centre (BioBank), Birmingham, for fresh tissue samples and anonymized clinical data. We thank Genomics Birmingham for sequencing. This work was supported by the Wellcome Trust Investigator Award (106955) to F.M., Cancer Research UK, Advanced Clinician Scientist Award (ref. C31641/A23923) to A.B. The funders had no role in study design, data collection and analysis, decision to publish or preparation of the manuscript. We would also like to thank C. Nepal and B. Lenhard for their comments on the manuscript.

## Author contributions

J.W., A.B. and F.M. conceived and coordinated the project. K.W., A.S., P.W., L.T. and J.S. generated and maintained CRC organoid lines. J.W. and Y.H. generated CAGE libraries. J.W. and P.W. carried out all other experiments and wet lab-based analyses. J.W. and Y.H. analyzed sequencing data. J.W. and F.M. interpreted the results with critical comments from A.B. and Y.H. J.W. and F.M. wrote the manuscript with support from A.B.

## Competing interests

The authors declare no competing interests.

## Additional information

**Extended data** is available for this paper at https://doi.org/10.1038/s41594-023-01156-8.

**Correspondence and requests for materials** should be addressed to Joseph W. Wragg, Andrew D. Beggs or Ferenc Müller.

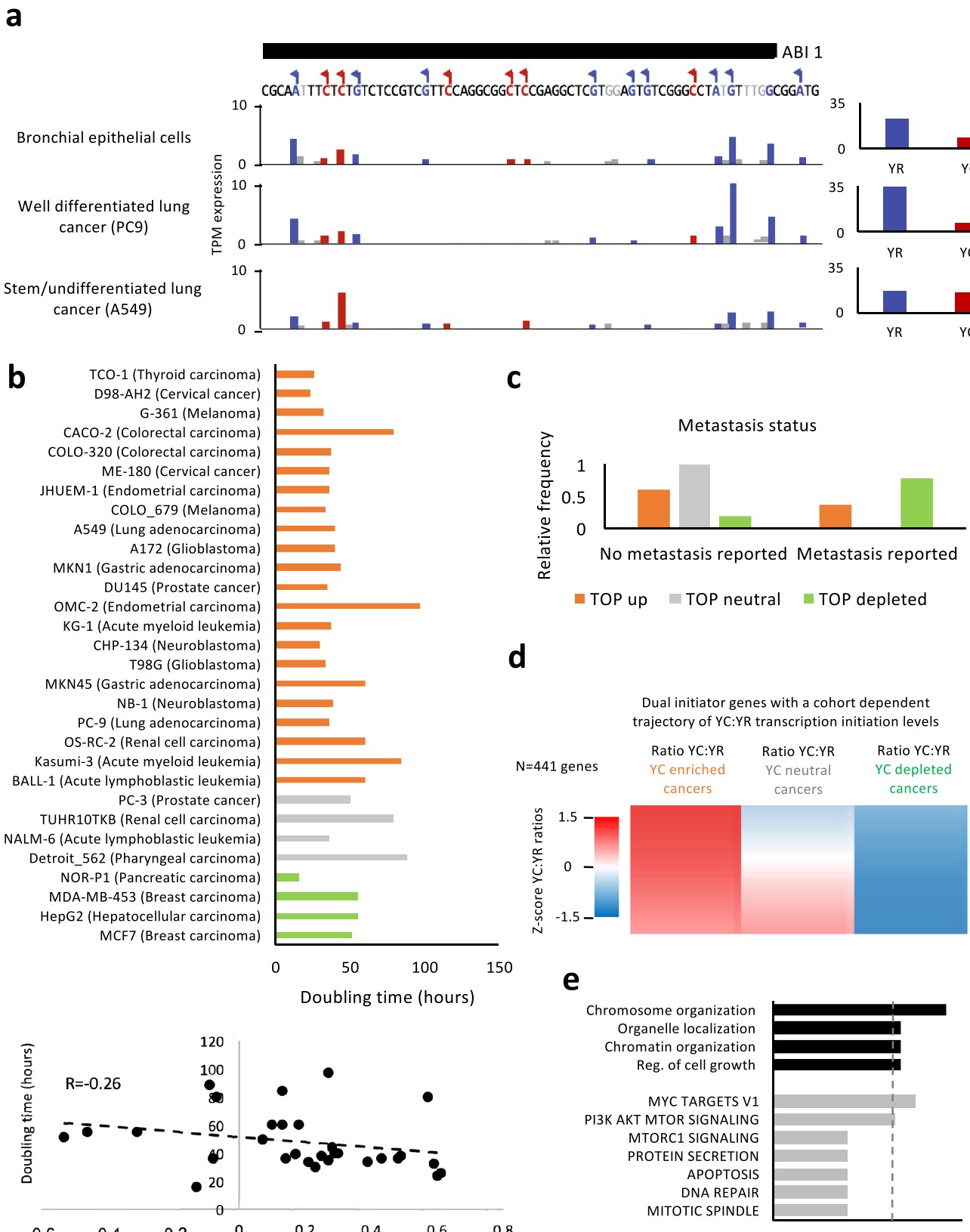

**Extended Data Fig. 1 | See next page for caption.**

**Extended Data Fig. 1 | YC transcription is most enriched in poorly differentiated and proliferative cancer types. a**, UCSC Genome Browser view of representative dual initiator gene, Abelson interactor 1 (ABI1), showing the relative usage of YC (red bars) and YR (blue bars) between healthy bronchial epithelial cells, a well differentiated lung cancer cell line (PC9) and a undifferentiated lung cancer cell line (A549), with total TPM values for each shown in bar graphs (right). This serves to exemplify the enhanced usage of YC transcription in undifferentiated cancer types. **b**, Top, bar graph of doubling times for the cancer cell lines analysed in Fig. 2, ordered by mean log2FC YC:YR transcription (cancer / Healthy) to match Fig. 2a. Bottom, scatter plot of correlation between mean log2FC YC:YR transcription (cancer / Healthy) and

cell line doubling time. **c**, Bar graph of the relative frequency of YC enriched, neutral and depleted cancer samples sourced from patients with/ without known metastasis. **d**, Dual initiator genes where the ratio of YC:YR transcription initiation dynamically changes between YC enriched, neutral and depleted cancer cohorts were identified (n = 422 promoters). A heatmap of the Z-score of YC:YR transcription ratios between cohorts is shown for this gene set **(d). e**, Bar graph of the gene ontology of dual initiators displaying dynamic YC:YR ratios between cohorts (as described in **d**). Significant biological process ontology (top) and match to Molecular signature database (MSigDB) Hallmark gene sets (bottom) are shown. Dotted line shows 0.05 FDR threshold.

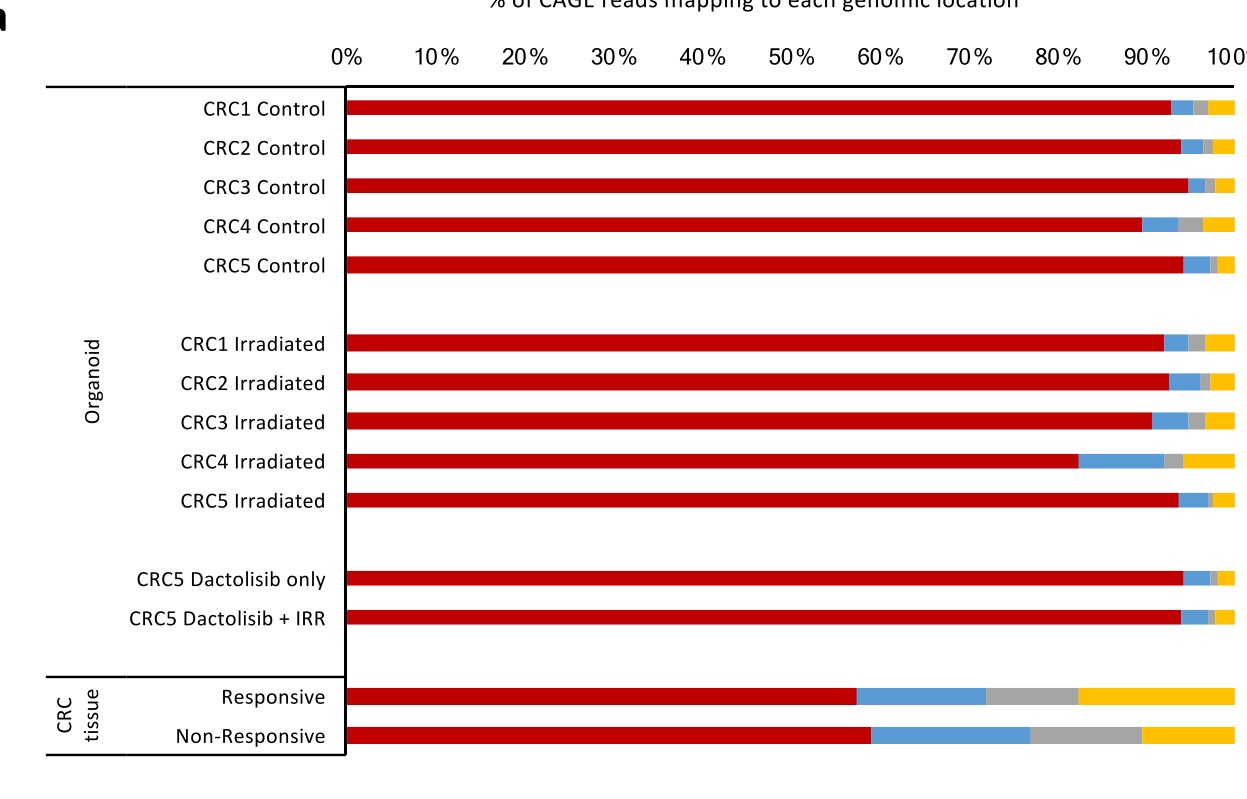

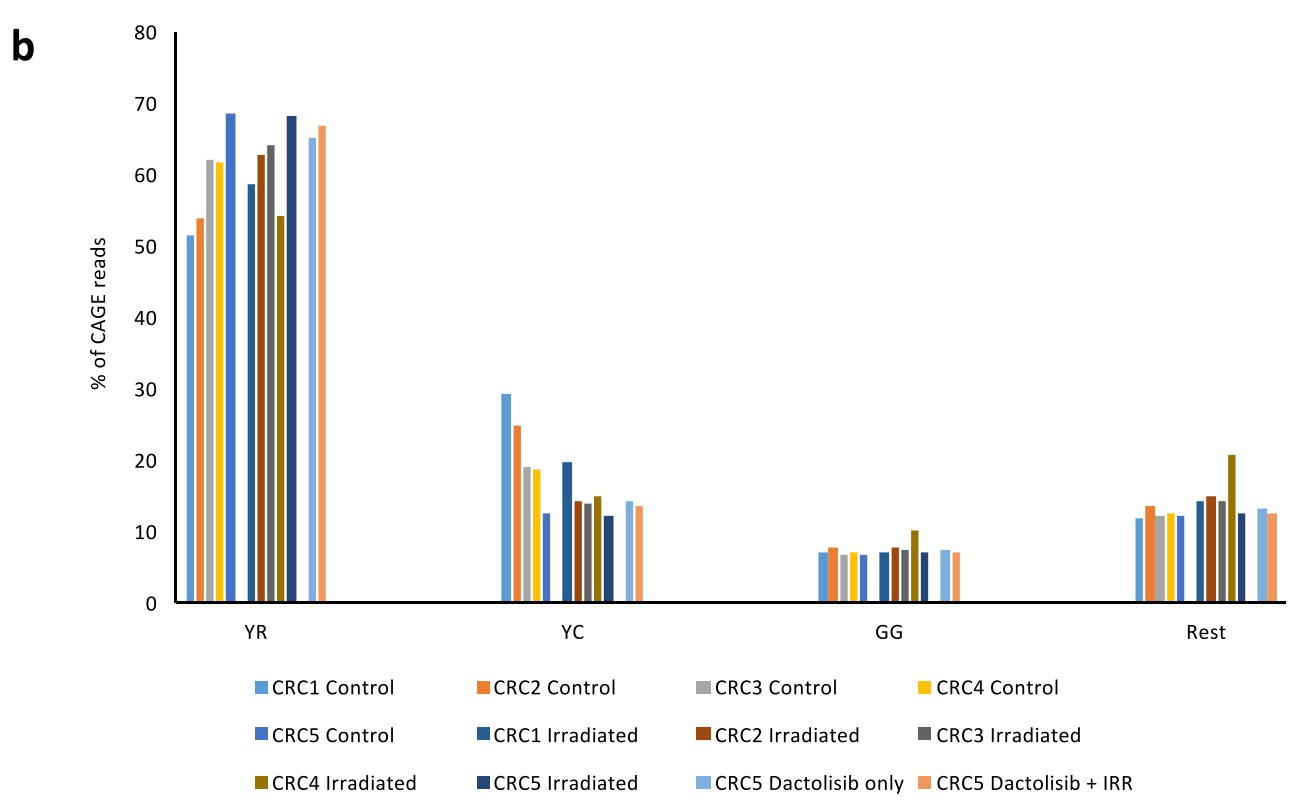

**Extended Data Fig. 2 | Quality assessment of new CAGE-seq datasets generated for this paper. a**, Bar graph of the frequency of CAGE read mapping to gene promoters, exons, introns and intergenic regions in each CAGE library.

**b**, Bar graph of the frequency of CAGE reads initiating at their 5' end with YR, YC, GG or other dinucleotides at the +1/−1 position respectively, for each CAGE library.

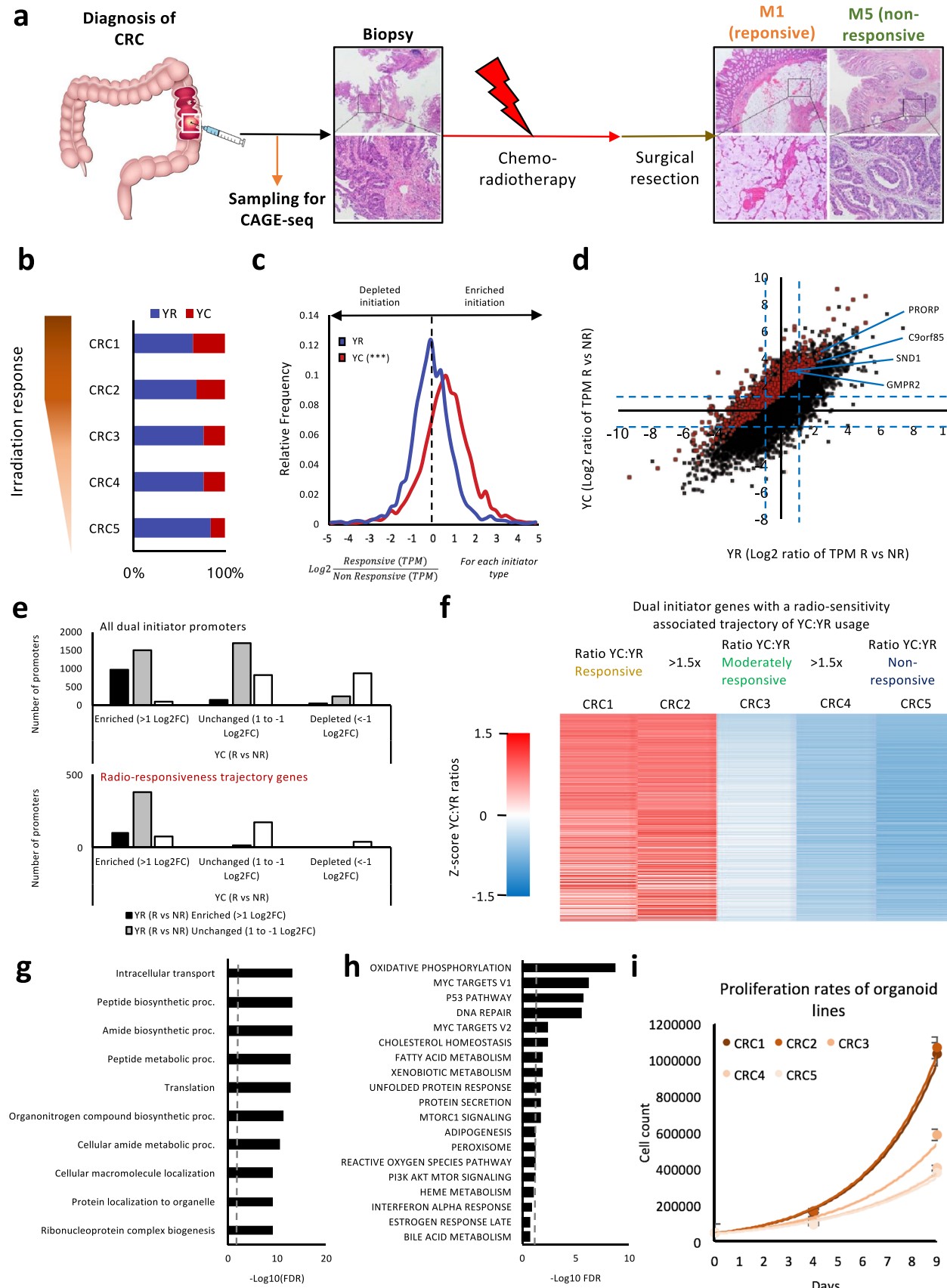

**Extended Data Fig. 3 | See next page for caption.**

**Extended Data Fig. 3 | Enriched YC initiation marks radiotherapy responsive CRC tumours. a**, Summary of the protocol for the collection of radio-responsive and resistant CRC tumour samples. Tumour images used with permission from[40]. **b**, Bar graph of the total expression from all CTSSs within consensus clusters (n = 18713), initiating with YR or YC dinucleotides, for each CRC organoid sample. **c**, Dual initiating promoters in the CRC organoid dataset were identified as previously described (n = 6285). The proportion of transcription initiating in each dual initiator promoter, from the YC and YR site was quantified for each sample and compared between them on a per promoter basis. Frequency distribution graph showing the degree of expression change of the YC (red) and YR (blue) component each dual promoter between responsive (average CRC1&2) and non-responsive (average CRC4&5) organoid samples (*** P < 0.001, T-test). **d**, Correlation scatter plot showing the relative expression of YC and YR components of all dual initiator genes (black) and Responsiveness trajectory genes (red) between responsive and non-responsive organoids (avr. CRC1&2 vs, avr. CRC4&5). Blue dotted lines show intersection with 1/−1 Log2FC. The

selected Radio-responsiveness signature genes explored by RTqPCR (Extended Data Fig. 8) are highlighted in this plot. **e**, bar graphs of the relative frequency of dual initiators displaying the behaviour of YR where YC is enriched / unchanged / depleted between responsive vs. non-responsive organoids. **f**, Dual initiator promoters with a dynamic shift in YC vs YR transcript abundance, correlating with radiotherapy responsiveness were identified (n = 807). The criteria used was that the average YC:YR ratio in CRC1&2 (responsive) for each dual initiator was 1.5 fold greater than the ratio in CRC3 (moderately responsive), which was in turn 1.5 fold greater than the YC:YR ratio in CRC4 and 5 (non-responsive). A heatmap of the relative YC:YR ratios for each dual initiator between CRC organoid samples is shown. **g**, **h**, Bar graphs of biological process (**g**) and Molecular signature database (MSigDB) Hallmark (**h**) gene ontology of dual initiators displaying dynamic YC:YR ratios correlated with radiotherapy sensitivity. Dotted line shows 0.05 FDR threshold. **i**, Line plot of organoid cell proliferation rate over 9 days, with cell counts taken at day 4 and day 9 (n = 3 independent experiments, data are presented as mean values +/- SEM).

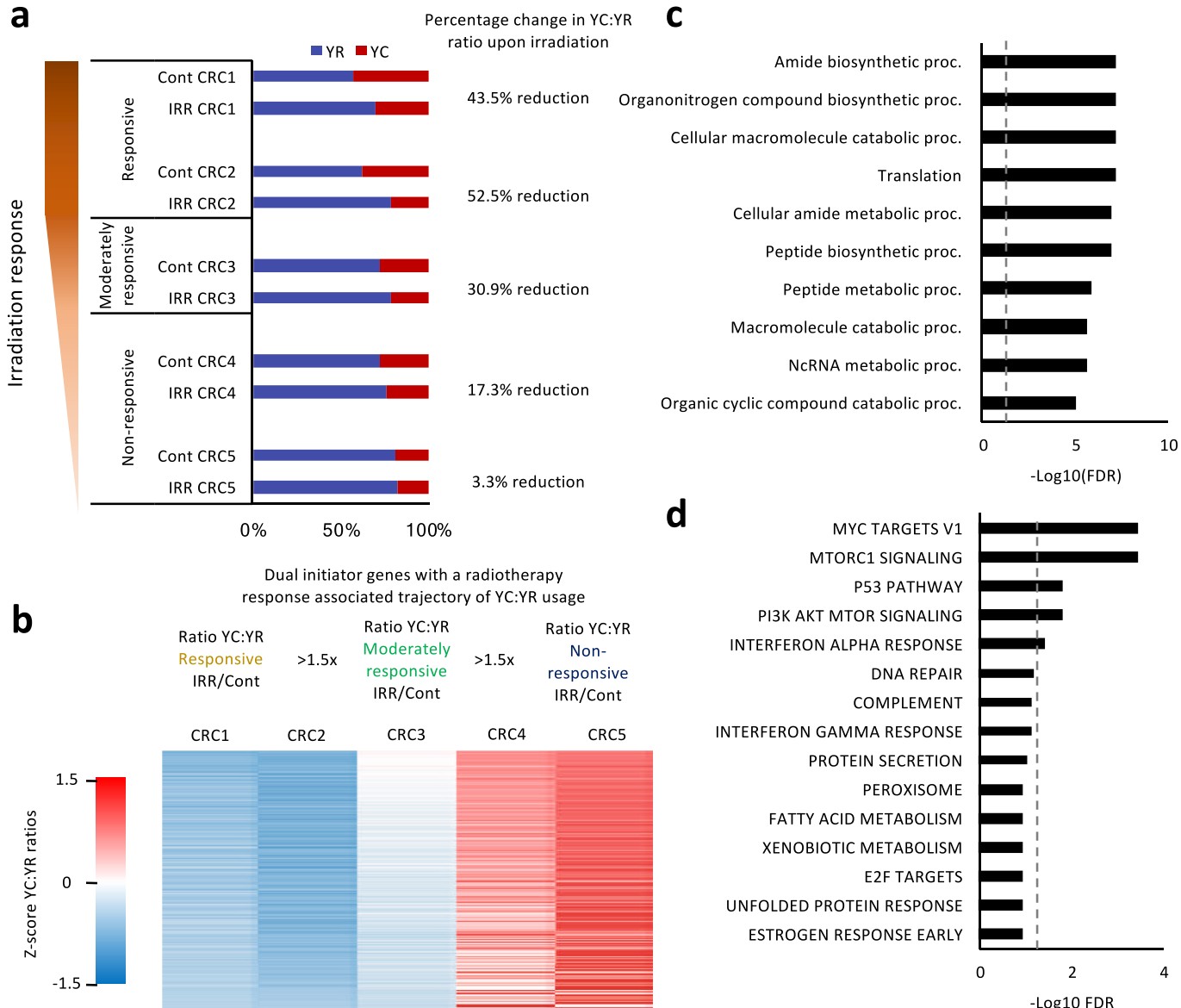

**Extended Data Fig. 4 | Radiotherapy responsive modulation of YC transcription initiation correlates with CRC clinical response. a**, Expanded version of Fig. 4a, illustrating the dynamics of total YC/YR TPM values upon irradiation for all 5 organoid samples. **b**, Dual initiator promoters with a dynamic shift in YC vs YR transcript abundance, upon irradiation, correlating with radiotherapy responsiveness (as illustrated in Fig. 4c) were identified (n = 411). The criteria used was that the average fold change in YC:YR ratio upon irradiation in CRC1&2 (responsive) for each dual initiator was 1.5 fold greater than the ratio in CRC3 (moderately responsive), which was in turn 1.5 fold greater than the YC:YR

ratio fold change upon irradiation in CRC4 and 5 (non-responsive). A heatmap of the relative fold change in YC:YR ratio upon irradiation for each dual initiator, between CRC organoid samples is shown. **c**, Bar graph of biological process gene ontology of dual initiators displaying dynamic YC:YR ratio change upon irradiation correlated with radiotherapy responsiveness (as described in **b**). **c**, Bar graph of the gene ontology of dual initiators displaying dynamic YC:YR ratios correlated with radiotherapy responsiveness (as described in **b**). Matches to the Molecular signature database (MSigDB) Hallmark gene sets (bottom) are shown. Dotted line shows 0.05 FDR threshold.

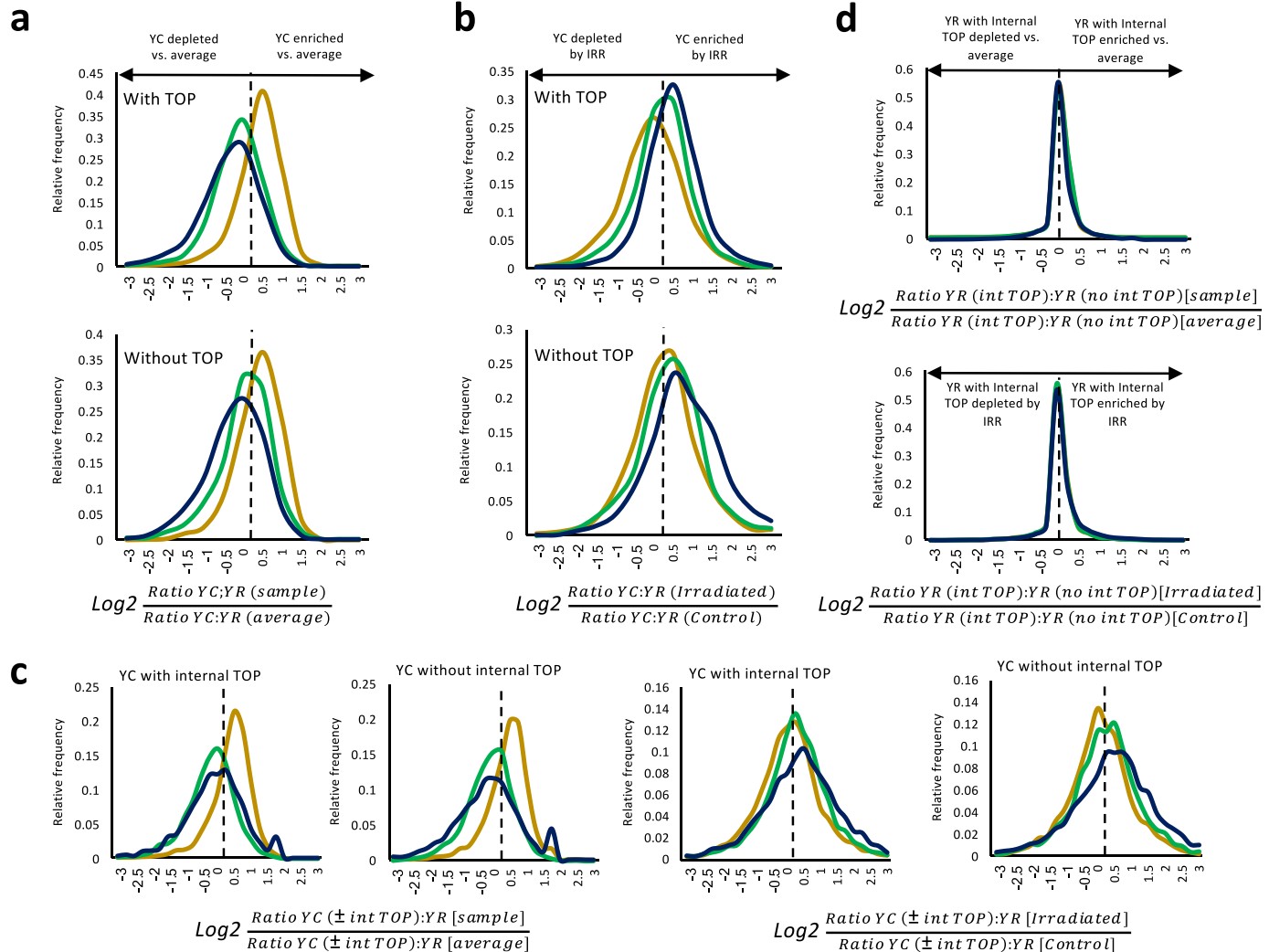

**Extended Data Fig. 5 | Enriched YC initiation marks radiotherapy responsive CRC tumours. a**, Dual initiators were segregated on the basis of whether they contained >1TPM of either TOP or TOP-deg YC forms (With TOP, n = 4819), or not (Without TOP – representing the group of 1466 DIPs with only the YC-other form identified in Fig. 5b). **a** shows the frequency distribution equivalent to Fig. 3g, but with the 'With TOP' and 'Without TOP' DI groups shown separately. **b**, frequency distribution equivalent to Fig. 4b, but with the 'With TOP' and 'Without TOP' DI groups shown separately. **c**, frequency distribution plots (analogous to **a**&**b**) of

the relative ratio of YC:YR transcripts where the YC transcripts are with or without internal TOP sequences (within the first 50 bp) in dual initiators in responsive (gold), moderately-responsive (green) and non-responsive (blue) organoid cohorts. **d**, frequency distribution plots (analogous to c&d) of the relative activity of YR transcripts with vs. without internal TOP sequences (within the first 50 bp) in consensus clusters in responsive (gold), moderately-responsive (green) and non-responsive (blue) organoid cohorts.

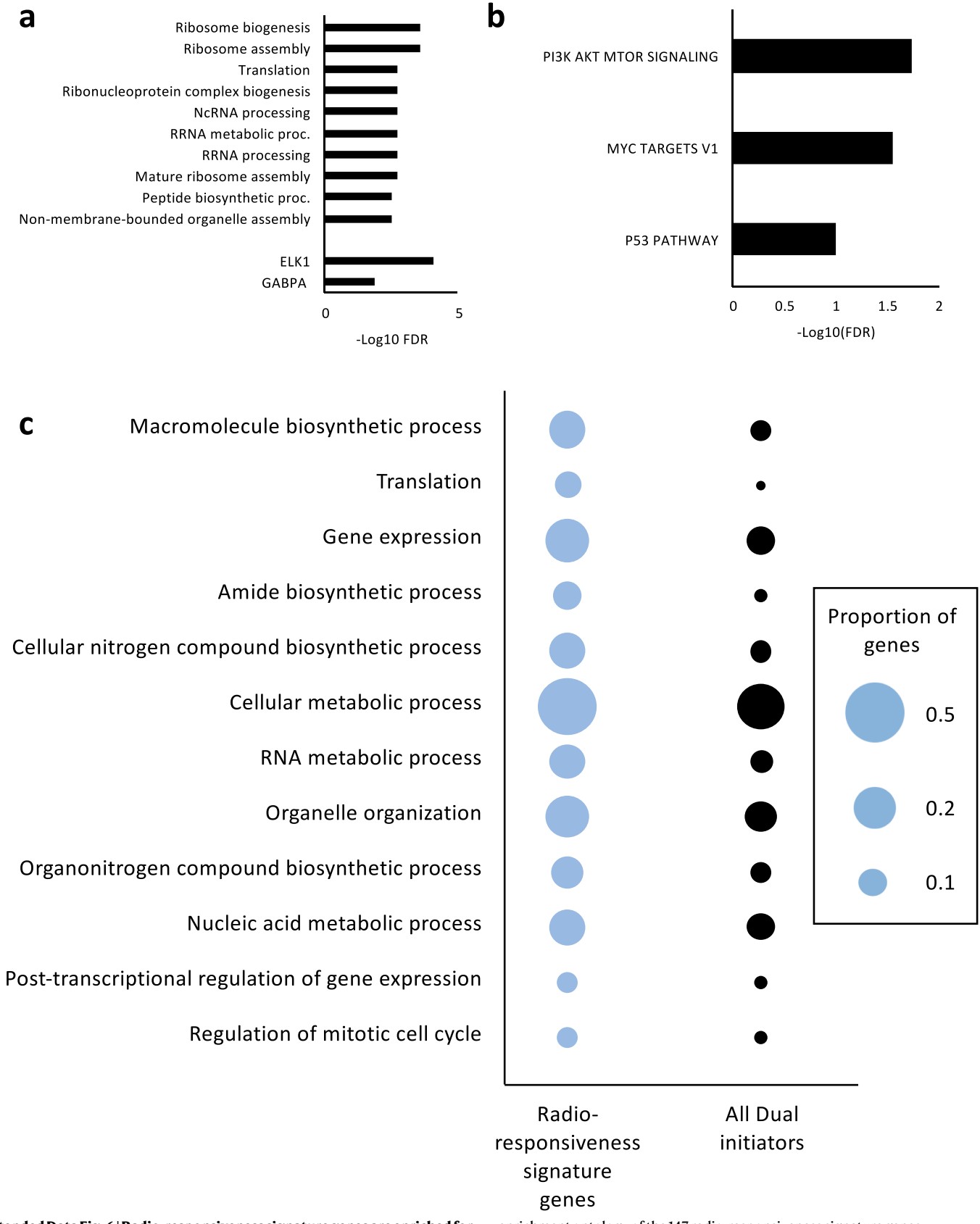

**Extended Data Fig. 6 | Radio-responsiveness signature genes are enriched for ribosomal and translation associated factors, but also represent a range of biological functions. a**, Bar graph of biological process gene ontology (top) and enriched motifs in the gene promoter (bottom) of the 147 radio-responsiveness signature genes (selected as described in Fig. 6). **b**, Bar graph of MSigDB pathway enrichment ontology of the 147 radio-responsiveness signature genes. **c**, Bubble graph displaying biological functions enriched in radio-responsiveness signature genes and ordered by P-value, but also represented by at least 10 genes in the radio-responsiveness signature gene set, to reveal the composition of the signature gene list.

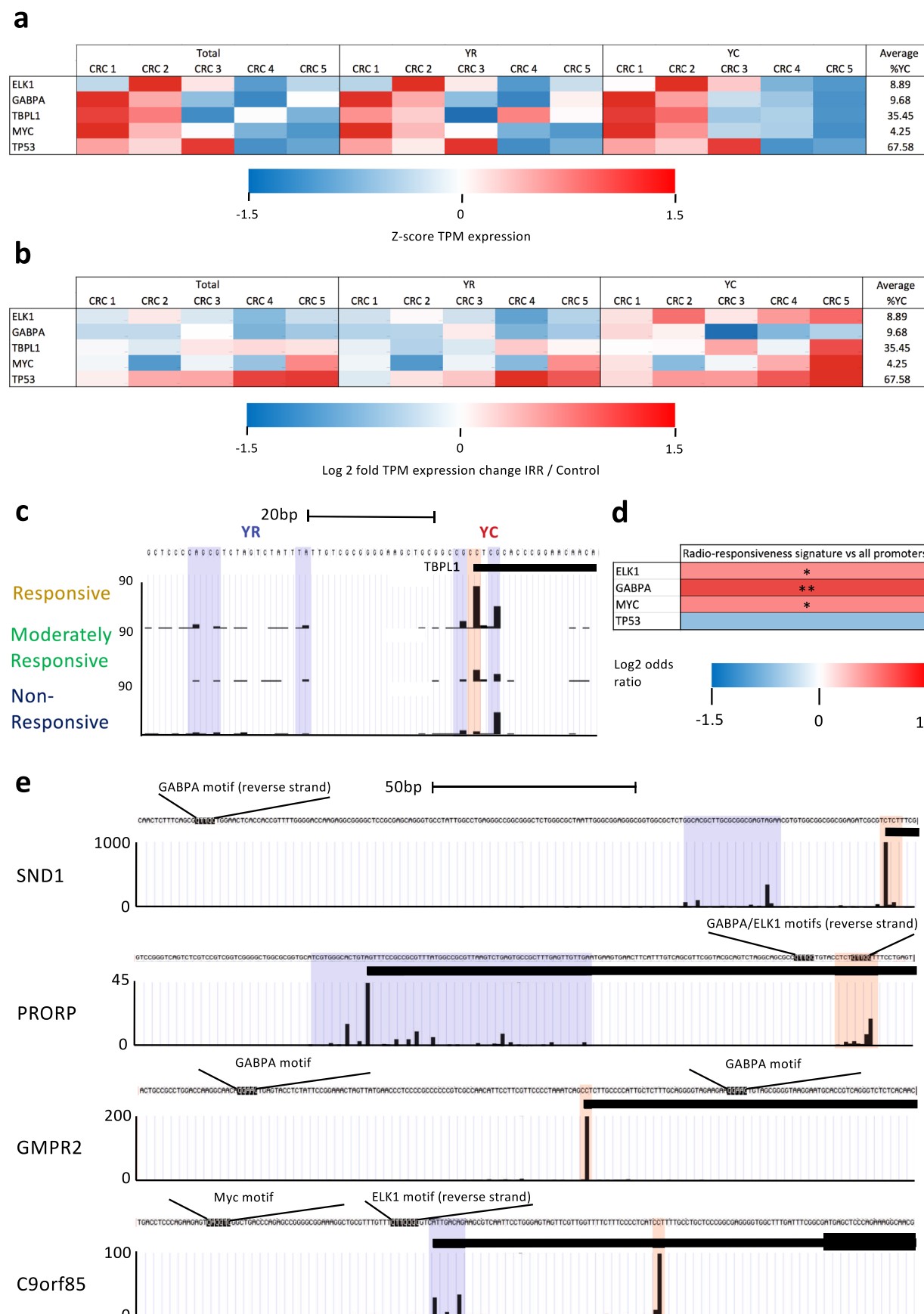

**Extended Data Fig. 7 | See next page for caption.**

**Extended Data Fig. 7 | MYC, ELK1 and GABPA transcription factor binding sites are enriched in the promoters of radiotherapy response signature genes. a**, Heatmap visualizing the Total, YR and YC transcript component expression patterns of transcription factors implicated in regulating the radio-responsiveness signature genes, across the 5 organoid samples. **b**, Heatmap visualizing the Total, YR and YC transcript component log2 fold change in expression of transcription factors implicated in regulating the radio-responsiveness signature genes, between irradiated and control samples of the 5 organoids. **c**, UCSC Genome Browser view of CAGE tracks from the transcription factor TBPL1, showing a dynamic switch from YC predominant transcription in radiotherapy responsive CRC organoids (CRC1) to YR predominant transcription in radiotherapy non-responsive organoids (CRC5), with balanced transcriptional output from YR and YC components in the moderately responsive CRC

organoid (CRC3). **d**, Heatmap visualizing the log2 odds ratio of the occurrence of core promoter motifs ( > 90% match to JASPAR published consensus motif, ELK1: MA0028.2, GABPA: MA0062.1, MYC: MA0147.3, TP53: MA0106.1) in the promoters (200 bp up and down stream of the dominant TSS) of the radio-responsiveness signature gene set vs. all promoters (P = 0.031, 0.002, 0.049 and 0.25 for ELK1, GABPA, MYC and TP53 respectively, two-tailed Fisher's exact test). **e**, Genome browser views of the candidate genes from the radio-responsiveness signature gene set with the location of proximal TF motifs highlighted. Reads from the responsive CRC cohort are shown in each case and the YR and YC transcriptional regions of the promoter highlighted in blue or red respectively. As the YC and YR transcription initiation sites in GMPR2 are spatially separated only the YC section is shown, however assessment of the YR region revealed no binding motifs corresponding to ELK1, GABPA, MYC or TP53.

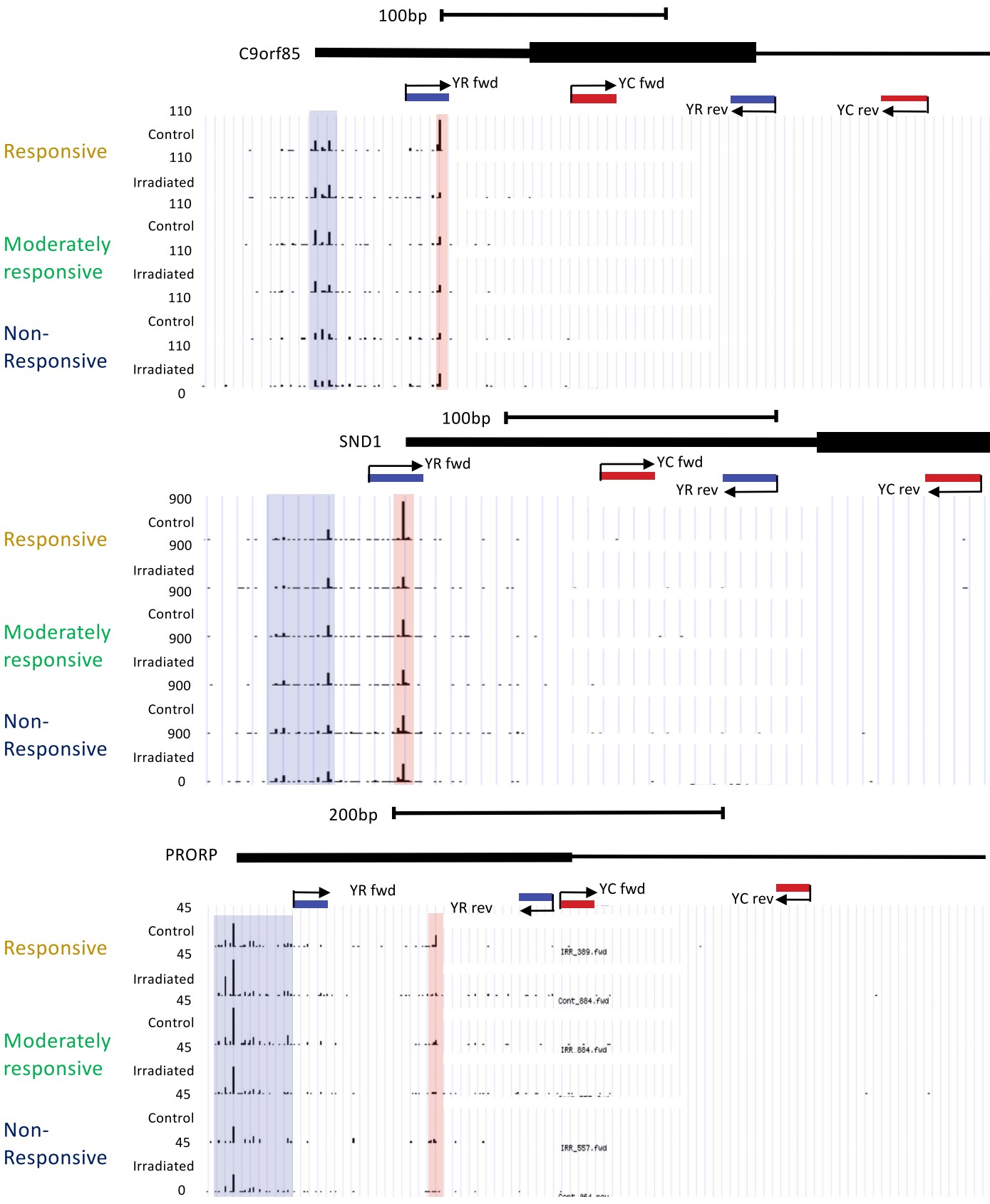

**Extended Data Fig. 8 | RT-qPCR Primer locations on candidate dual initiator promoters.** UCSC Genome Browser views of dual initiating promoters, highlighting the location of annealing sites for primers designed to segregate the expression of the YC and YR component through RTqPCR analysis. The YR and YC initiation regions of the promoter are denoted by blue and red boxes respectively.

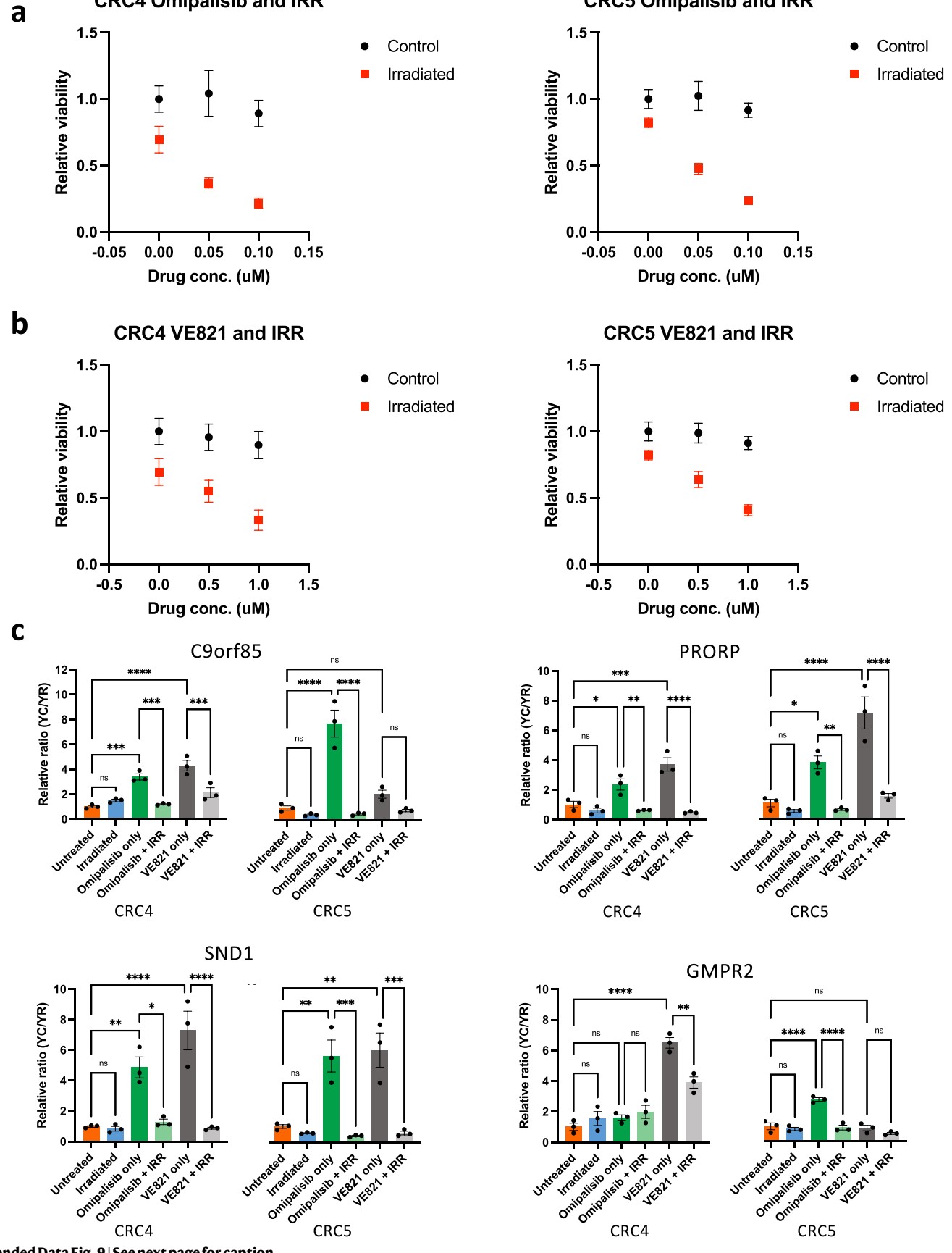

**Extended Data Fig. 9 | See next page for caption.**

**Extended Data Fig. 9 | Inhibition of PI3K / AKT / mTOR and DNA damage pathway signalling enhances YC transcript abundance and restores radiotherapy induced transcriptional dynamics. a** and **b**, Plot of survival of organoids from each resistant line, exposed to a combination of drug treatment (**a** – Omipalisib, **b** – VE821) and irradiation (Data are presented as mean values +/- SEM). **c**, Bar graphs of RTqPCR analysis of relative YC/YR expression from candidate dual initiators in CRC organoids treated with Omipalisib and VE821, irradiation or a combination of drug and irradiation (n = 3 independent experiments, * P < 0.05, ** P < 0.01, *** P < 0.001, **** P < 0.0001, Ordinary one-way ANOVA with Tukey's multiple comparison's test, Data are presented as mean values +/- SEM, full list of p-values available presented in the Source data file).

# Reporting Summary

## Statistics

For all statistical analyses, confirm that the following items are present in the figure legend, table legend, main text, or Methods section.

| n/a | Confirmed | |
|---|---|---|
| ☐ | ☒ | The exact sample size (*n*) for each experimental group/condition, given as a discrete number and unit of measurement |
| ☐ | ☒ | A statement on whether measurements were taken from distinct samples or whether the same sample was measured repeatedly |
| ☐ | ☒ | The statistical test(s) used AND whether they are one- or two-sided <br> *Only common tests should be described solely by name; describe more complex techniques in the Methods section.* |
| ☒ | ☐ | A description of all covariates tested |
| ☐ | ☒ | A description of any assumptions or corrections, such as tests of normality and adjustment for multiple comparisons |
| ☐ | ☒ | A full description of the statistical parameters including central tendency (e.g. means) or other basic estimates (e.g. regression coefficient) AND variation (e.g. standard deviation) or associated estimates of uncertainty (e.g. confidence intervals) |
| ☐ | ☒ | For null hypothesis testing, the test statistic (e.g. *F*, *t*, *r*) with confidence intervals, effect sizes, degrees of freedom and *P* value noted <br> *Give P values as exact values whenever suitable.* |
| ☒ | ☐ | For Bayesian analysis, information on the choice of priors and Markov chain Monte Carlo settings |
| ☐ | ☒ | For hierarchical and complex designs, identification of the appropriate level for tests and full reporting of outcomes |
| ☐ | ☒ | Estimates of effect sizes (e.g. Cohen's *d*, Pearson's *r*), indicating how they were calculated |

*Our web collection on statistics for biologists contains articles on many of the points above.*

## Software and code

Policy information about availability of computer code

| Data collection | N/A |
|---|---|
| Data analysis | Data was analyzed using publicly available software, which are cited in the paper. The version of software and syntax used are listed in the paper. |

For manuscripts utilizing custom algorithms or software that are central to the research but not yet described in published literature, software must be made available to editors and reviewers. We strongly encourage code deposition in a community repository (e.g. GitHub). See the Nature Portfolio guidelines for submitting code & software for further information.

## Data

Policy information about availability of data

All manuscripts must include a data availability statement. This statement should provide the following information, where applicable:
- Accession codes, unique identifiers, or web links for publicly available datasets
- A description of any restrictions on data availability
- For clinical datasets or third party data, please ensure that the statement adheres to our policy

All relevant data and results included in this article have been published along with the article and its supplementary information files. Raw sequencing data for CAGE-seq is publicly available at NCBI Sequence Read Archive under accession number PRJNA934878. Public data from FANTOM5, was downloaded from https://fantom.gsc.riken.jp/5/datafiles/latest/basic/.

# Human research participants

Policy information about studies involving human research participants and Sex and Gender in Research.

| | |
|---|---|
| Reporting on sex and gender | Sex/gender information, where available, is provided in Supplementary table 2. As described in the paper it was assessed, but not found to correlate with transcription initiation behavior. Sex/gender information was not collected under the ethics for the organoid or clinical samples, and therefore was not considered. |
| Population characteristics | Population characteristics are provided in Supplementary tables 2 and 4. All available characteristics were assessed with their relevance discussed in the manuscript. |
| Recruitment | No human participants were recruited for this study as it is pre-clinical in nature, with only anonymised archived tissues or established cultures used. |
| Ethics oversight | Project approval code 17-287, Human Biomaterials Resource Centre, Birmingham, United Kingdom; under ethical approval from Northwest - Haydock Research Ethics Committee (Reference: 15/NW/0079). |

Note that full information on the approval of the study protocol must also be provided in the manuscript.

# Field-specific reporting

Please select the one below that is the best fit for your research. If you are not sure, read the appropriate sections before making your selection.

☒ Life sciences  ☐ Behavioural & social sciences  ☐ Ecological, evolutionary & environmental sciences

For a reference copy of the document with all sections, see nature.com/documents/nr-reporting-summary-flat.pdf

# Life sciences study design

All studies must disclose on these points even when the disclosure is negative.

| | |
|---|---|
| Sample size | No statistical method was used to predetermine sample size as this was rather determined by maximum sample availability which was limited in the case of organoids by numbers successfully generated, clinical samples by ethics agreements and previously published FANTOM5 CAGE data by cancer samples having suitable match healthy tissue samples. |
| Data exclusions | No data were excluded from the analyses. |
| Replication | Experimental data was the product of 3 independent experiments unless otherwise stated. |
| Randomization | The experiments were not randomized as this this was not relevant to the study. All analyses were objective in nature and not the product of a survey or subjective scoring approach. |
| Blinding | Investigators were not blinded to allocation during experiments and outcome assessment as this this was not relevant to the study. All analyses were objective in nature and not the product of a survey or subjective scoring approach. The only subjective analysis was the organoid morphology scoring in figure 3h, but with a sample size of 5, blinding was not practicable and images of each organoid are shown for the reader to make their own judgment. |

# Reporting for specific materials, systems and methods

We require information from authors about some types of materials, experimental systems and methods used in many studies. Here, indicate whether each material, system or method listed is relevant to your study. If you are not sure if a list item applies to your research, read the appropriate section before selecting a response.

## Materials & experimental systems

| n/a | Involved in the study |
|---|---|
| ☒ | ☐ Antibodies |
| ☒ | ☐ Eukaryotic cell lines |
| ☒ | ☐ Palaeontology and archaeology |
| ☒ | ☐ Animals and other organisms |
| ☒ | ☐ Clinical data |
| ☒ | ☐ Dual use research of concern |

## Methods

| n/a | Involved in the study |
|---|---|
| ☒ | ☐ ChIP-seq |
| ☒ | ☐ Flow cytometry |
| ☒ | ☐ MRI-based neuroimaging |

