## [Peer Review File · Nature Structural & Molecular Biology]

Peer Review Information

Manuscript Title: Intra-promoter switch of transcription initiation sites in proliferation signalling-dependent RNA metabolism

Corresponding author name(s): Joseph Wragg, Andrew Beggs, Ferenc Müller

Reviewer Comments & Decisions:

Decision Letter, initial version:

Message: 20th Mar 2023

Dear Dr. Wragg,

Thank you again for submitting your manuscript "Transcription initiation site choice precedes mTOR-dependent RNA metabolism and defines cancer differentiation and radiotherapy sensitivity". I apologise for the delay in responding, which resulted from the difficulty in obtaining suitable referee reports. Nevertheless, we now have comments (below) from the 3 reviewers who evaluated your paper. In light of those reports, we remain interested in your study and would like to see your response to the comments of the referees, in the form of a revised manuscript.

You will see that all reviewers appreciate the conceptual novelty of the findings, the support provided for drawn conclusions, and the potential mechanistic/functional implications of the proposed hypotheses. However, there are important technical concerns that need to be addressed and a few experimental suggestions that should be included in a revised manuscript. More specifically, all three reviewers point out technical ambiguities (for example see points 1, 2, 5, 6 of Reviewer #1, points 2,3 of Reviewer #3) and the need for expanding current analyses to better control demonstrated data compared to healthy background (for example see points 1,2 of Reviewer #2 and 4 of #3) and expand their implications (for example see points 1-2 of R#1 on the potential connection between p53 status or proliferation rate and YC usage, point 5 of R#1 about building an mTORC1-signature of overlapping genes instead of GO terms, point 5 of R#2). Moreover, the reviewers request a few experimental suggestions that we editorially agree will significantly strengthen the mechanistic and functional findings and implications of this work. Namely, R#1 requests that you validate the hypothesis that PI3K/mTOR increases YC usage and radio-sensitivity with more specific mTOR inhibitors, that do not additionally inhibit the master DNA damage response kinases, ATR and ATM (point 7). Finally, R#3 requests generating a dual-initiation sequence promoter reporter and validating some of the candidate genes (point 1).

Please be sure to address/respond to all concerns of the referees in full in a point-by-point response and highlight all changes in the revised manuscript text file. If you have comments that are intended for editors only, please include those in a separate cover letter.

We expect to see your revised manuscript within 3 months. If you cannot send it within this time, please contact us to discuss an extension; we would still consider your revision, provided that no similar work has been accepted for publication at NSMB or published elsewhere.

Reporting Summary:

When submitting the revised version of your manuscript, please pay close attention to our [href="https://www.nature.com/nature-portfolio/editorial-policies/image-integrity">Digital Image Integrity Guidelines. and to the following points below:](https://www.nature.com/nature-portfolio/editorial-policies/image-integrity)

Please note that all key data shown in the main figures as cropped gels or blots should be presented in uncropped form, with molecular weight markers. These data can be aggregated into a single supplementary figure item. While these data can be displayed in a relatively informal style, they must refer back to the relevant figures. These data should be submitted with the final revision, as source data, prior to acceptance, but you may want to start putting it together at this point.

SOURCE DATA: we urge authors to provide, in tabular form, the data underlying the graphical representations used in figures. This is to further increase transparency in data reporting, as detailed in this editorial

(<http://www.nature.com/nsmb/journal/v22/n10/full/nsmb.3110.html>). Spreadsheets can be submitted in excel format. Only one (1) file per figure is permitted; thus, for multi-paneled figures, the source data for each panel should be clearly labeled in the Excel file; alternately the data can be provided as multiple, clearly labeled sheets in an Excel file. When submitting files, the title field should indicate which figure the source data pertains to. We encourage our authors to provide source data at the revision stage, so that they are part of the peer-review process.

Data availability: this journal strongly supports public availability of data. All data used in accepted papers should be available via a public data repository, or alternatively, as Supplementary Information. If data can only be shared on request, please explain why in your Data Availability Statement, and also in the correspondence with your editor. Please note that for some data types, deposition in a public repository is mandatory - more information on our data deposition policies and available repositories can be found below: <https://www.nature.com/nature-research/editorial-policies/reporting-standards#availability-of-data>

Deposition of deep sequencing and microarray data is mandatory, and the datasets must be released prior to or upon publication. To avoid delays in publication, dataset accession numbers must be supplied with the final accepted manuscript and appropriate release dates must be indicated at the galley proof stage.

Nature Structural & Molecular Biology is committed to improving transparency in authorship. As part of our efforts in this direction, we are now requesting that all authors identified as 'corresponding author' on published papers create and link their Open Researcher and Contributor Identifier (ORCID) with their account on the Manuscript Tracking System (MTS), prior to acceptance. This applies to primary research papers only. ORCID helps the scientific community achieve unambiguous attribution of all scholarly contributions. You can create and link your ORCID from the home page of the MTS by clicking on 'Modify my Springer Nature account'. For more information please visit please

visit <http://www.springernature.com/orcid>

[redacted]

Sincerely,

Dimitris Typas
Associate Editor
Nature Structural & Molecular Biology
ORCID: 0000-0002-8737-1319

Referee expertise:

Referee #1: mTOR signaling and transcription (5')/translational control

Referee #2: Computational cancer genomics, CAGE-Seq

Referee #3: colorectal cancer genetics, stem cells, organoid culture

Reviewers' Comments:

Reviewer #1:

Remarks to the Author:

This manuscript from Wragg et al. investigates patterns of transcription start site (TSS) selection that distinguish cancer cells from normal cells. This builds on previous work from this group that profiled the use of canonical YR TSSs and non-canonical YC TSSs across metazoan promoters. Here, the authors surprisingly find that YC usage preferentially increases in many cancer cell lines. This phenomenon correlates with the differentiation state of the cells, where decreased differentiation increases YC usage. The authors find that increased YC usage also correlates with sensitivity to radiotherapy in primary colorectal tumors and organoids. From these results, they define a signature of 4 genes with distinct YC/YR usage to distinguish radiosensitive from resistant tumors that may have prognostic value. Finally, the authors argue that mTOR/PI3K inhibition, which has previously been shown to increase radiosensitivity, also increases YC usage in their organoid model, suggesting that these phenomena may be connected.

The main conclusion of this work is that YC/YR usage reflects a global transcriptional regulatory mechanism that correlates with cell states that are relevant to cancer growth and treatment. This is an exciting and novel concept that prompts many additional questions about the nature of the underlying mechanism. The impact of this message, however, is weakened by frequently confusing data presentation, unclear methodology, and some experimental concerns. This reviewer is enthusiastic about the main message of this manuscript and hope that the authors will be able to address these concerns. Details are provided below.

Major points:

1. A central finding is that YC usage increases in cancer cells categorized as "less differentiated". A few clarifications are needed. First, the authors don't say how they define differentiation status. Is this from the literature? Morphology? Expression of a gene signature? This should at least be described better in the methods. Second, is it possible that proliferation rate is a stronger determinant of YC usage? For instance, Figure 2c

shows that YC-enriched cancers have elevated expression of many cell cycle gene categories. The authors also hint at a relationship between proliferation, undifferentiated status, and radiosensitivity in the text (lines 392-394). The authors should test, at least for a subset of cancer cell lines in their panel, the relationship between proliferation rate and YC usage. Third, do any mutations aside from p53 correlate with preferential YC usage? Mutation data for many of these cell lines is available from the Cancer Cell Line Encyclopedia (Broad Institute).

2. Similar to the above comment, more information is needed about the CRC organoids described first in Figure 3. How were these obtained? Do their proliferation rates correlate with IRR sensitivity? What is their p53 mutation status? The authors showed striking differences in IRR sensitivity amongst the 5 CRC organoid lines. As with the cancer cell lines analyzed in Figure 2, is it possible that proliferation rate is a better (and more quantitative) indicator of both YC usage and IRR sensitivity?

3. In the overall TSS analysis of organoids in Figure 3f, it appears that YC usage generally changes while YR usage remains roughly constant. This also seems to be true for SND1 (Figure 3h). Are there examples where YR usage changes while YC usage remains constant? Or does only YC usage change? It would be helpful to visualize this, perhaps by providing a scatterplot showing levels of YC and YR TSSs in sensitive versus responsive organoids?

4. Figure 4 shows that IRR decreases the YC/YR ratio in IRR-responsive organoids to levels similar to untreated non-responsive organoids. A simplistic interpretation would be that the decreased YC usage indicates a shift to a radio-resistant state, but this seems illogical. An alternative interpretation is that IRR selects for radio-resistant cells with low YC usage within organoids, and that the frequency of these cells is higher in resistant organoids. Could the authors comment on this possibility?

5. The manuscript relies heavily on GO categories of genes to make the point that YC usage increases in many of the same genes under different contexts (e.g. differentiation status, irradiation) (Figure 2f, 3j, 4e). The meaning of these categories is confusing to this reviewer. Do they contain overlapping sets of genes? What is included in MTORC1 Signaling? Is it possible to instead define a unified gene signature of dual-initiator genes with variable YC/YR ratios from results in Figure 2f, and then analyze and/or refine this gene set in later figures? This would clarify the specific genes that link differentiation status to IRR sensitivity. The authors should also provide the list of the 148 radio-responsiveness signature genes shown in Figure 5a as a supplemental table.

6. More information on the "delta-delta DE-Seq" analysis is needed. The approach is currently described only in the main text (lines 365-368), which appears to first normalize gene expression levels in cancer cells to levels in normal tissue. This is not appropriate for DESeq, which relies on raw read counts for the particular statistical model used to determine significance of differences in expression levels (Love et al., Genome Biol, PMID: 25516281). DESeq can handle ratios-of-ratios (e.g. YC versus YR for cancer versus normal) with the appropriate model design.

7. The authors use NVP-BEZ235 to test their hypothesis that PI3K/mTOR inhibition increases YC usage and radio-sensitivity. However, this molecule is also a strong inhibitor of ATM and ATR (Toledo et al., NSMB, PMID: 21552262), which may contribute more to the increased sensitivity to IRR than mTOR/PI3K inhibition. Furthermore, the increase in

YC transcript levels could also be explained by increased mRNA stability following BEZ235 treatment, since mTOR inhibition stabilizes YC mRNAs. To show that this effect is specific to the mTOR and/or PI3K pathway, these experiments would need to be repeated with more specific inhibitors. Many such compounds are available. The authors should also at least discuss the possibility that the increase in YC usage in inhibitor-treated cells may reflect changes in stability rather than transcription (and preferably show this directly using reporters).

8. The manuscript includes a lengthy discussion (lines 626-683) of whether TOP mRNAs, which could be considered extreme examples of YC-hi genes, drive the increase in YC usage in radio-sensitive samples. The conclusion is that increased TSS usage for TOP genes reflects a broader phenomenon that globally increases YC usage. As nearly all of the associated data is in the Supplemental section, it seems that this section could be significantly condensed. As written, it distracts from the main message that YC usage is globally increased through a mechanism that is not yet understood.

Minor points:

1. Figures 1b and 1c are conceptually redundant. Is it possible to combine them into a single panel?
2. How many tumor samples were analyzed in Figures 2a-d? 6 responsive and 6 non-responsive are listed in Supplemental Table 4, but Supplemental Table 5 suggests that only one from each category was analyzed by CAGE. Is that correct? More information is needed in the Methods section as to how these samples were obtained.
3. The heatmaps shown in Figures 2e, 3i, 4d seem unnecessary, as these gene sets have been selected based on a shared pattern of expression. I recommend that they be removed.
4. This sentence at lines 456-458 is unclear: "Additionally, the ability of the technique to identify dual initiator promoters appears to be considerably poorer than traditional CAGE used for the FANTOM5 cancer data (186 Dis [5.3% of promoters, FFPEcap-seq], 3475 Dis [19.9%])." What is "Dis"? 3475 is 19.9% of what?
5. Is the increased expression of YR transcripts in the "other promoter" category for CRC4 & 5 (Figure 3f) significant?
6. It would make more sense to show the qPCR analysis of clinical CRC samples and the analysis of CRC organoid samples as separate figure panels. The analysis of CRC organoids validates the ability of the qPCR method to distinguish YR and YC usage for these transcripts, but the correlation with radio-sensitivity is expected because these genes were identified based on results from these same samples. The analysis of clinical CRC samples is much more important because it validates the strategy for assessing samples that were not used to identify this gene signature. Moving these results to their own panel would strengthen this point.

Reviewer #2:

Remarks to the Author:

Wragg et al. report that the transcription initiation site usage switches from YR to YC in cancer samples and this switch is correlated with differentiation status of cancer types. They identified the enrichment of YC usage in dual initiators is related to radiotherapy responsive CRC and CRC organoid samples. By inhibiting the PI3K/Akt/mTOR pathway, CRC organoids insensitive to radiotherapy was shown to restore sensitivity. Overall, the promoter switch is a consequence of mTOR alternation which is measured by CAGE. The method and conclusion of the study are appropriate, and the content of this study is worth publishing. However, I have some questions and comments as follows:

1. Do all the CRC organoids show cancer phenotypes? I assume it can be extracted from the organoid CAGE data or other IF results easily. According to the first part of the study, lower YC/YR ratio may refer to more differentiated cancer or non-cancer. If radiotherapy select to kill more aggressive (undifferentiated) types but not the non-cancer cells, is it needed to manipulate the cells to be killed? On the other hand, if the Dacto treatment is applied to patients, will non-cancer cells be sensitized to radiotherapy? These should be discussed in the manuscript. If a healthy organoid is available, same experiment described in Fig. 6 will directly resolve this concern.
2. If data is available, it will be good to include non-cancer clinical sample / organoid from healthy subject in Fig. 3B & F, to observe if there is any YC/YR difference between non-responsive and normal.
3. What are the selection criteria for the 4 genes with qPCR results? For the least responsive gene, GMPR2, is it free of related TF binding motif? It would be interesting to see the CAGE-seq result derived from the Dacto-treated organoid to reveal the actual effect on specific dual initiators.
4. In Fig. 2F and Fig. 5B, non-significant pathways were cherry-picked to show. I don't think it is necessary and meaningful. And in line 699: "MYC, MTORC1 and P53 regulatory pathways (Figure 5B, Supplementary figure 5A), in agreement with Figure 2F, Figure 3J and Figure 4E." MYC is significant in 3 out of the 4, MTORC1 and P53 is significant in 2 out of the 4. Please revise this statement. Alternatively, test level analysis can be performed across the pathways to evaluate if MYC, MTORC1 and P53 are consistently enriched significantly in the 4 tests (10.1038/s41467-019-13983-9).
5. Downstream 5'TOP-like can be tested directly (line 662) from the YC-other. It would be good if the authors can perform the analysis and reveal this possibility.
6. For the TF binding motif analysis, please include available motif in JASPAR. And with the p-value adjusted into FDR. This can ensure the analytical pipeline is set correctly.

Reviewer #3:

Remarks to the Author:

Variations in transcription start site (TSS) selection reflect the diversity of preinitiation complexes and can impact post-transcriptional RNA fates, especially the recent identification of dual-initiation promoters genes carrying canonical initiation with pyrimidine/purine (YR) dinucleotide while carrying polypyrimidine initiator (5'-TOP) translation machinery-associated genes. However, the functional significance of start site choice in these dual promoter architectures is little understood. The authors first

systematically identified genes that mark YC enriched vs YC depleted in cancers. Enriched YC transcription correlated with poorly differentiated, proliferative (and potentially TP53 mutated) cancer subtypes led to a more specific examination of the developmental regulation of TSS selection in the chemo radio-resistance of the CRC. Overall, the hypothesis is novel, and the data also propose for the first time that dual initiation on shared promoters represents a composite promoter architecture in genes involved in cancer biology and diversity in treatment.

The manuscript is also written well, and however, I have some mid & minor concerns before considering for publication.

Mid/ Majors:

1. The author could construct a reporter vector with dual-initiation sequences/ promoter fused to luciferase (LacZ gene) and evaluate some candidate genes correlated with different stages of patient samples in CRC cell lines.
2. As radiation therapy is more associated with rectal cancer, it is beneficial for authors to define which of CRC 1-5 are rectal or colon-derived organoids.
3. The authors need to describe in more detail how and why they used 25Gys of irradiation over five days and their genetic background.
4. Also, authors could provide images of organoids from responsive, moderately responsive and non-responsive for their phenotypical characteristics.

Minors;

1. In figure 6a, the organoid images are the same as each other
2. If figure 6b, p-values are missing
3. A couple of sentences need a bit of rewriting, e.g., on Page 16, line 427- suggesting efficient start site detection and the that biological replicates of responsive

Author Rebuttal to Initial comments

Reviewer #1 (Remarks to the Author)

This manuscript from Wragg et al. investigates patterns of transcription start site (TSS) selection that distinguish cancer cells from normal cells. This builds on previous work from this group that profiled the use of canonical YR TSSs and non-canonical YC TSSs across metazoan promoters. Here, the authors surprisingly find that YC usage preferentially increases in many cancer cell lines. This phenomenon correlates with the differentiation state of the cells, where decreased differentiation increases YC usage. The authors find that increased YC usage also correlates with sensitivity to radiotherapy in primary colorectal tumors and organoids. From these results, they define a signature of 4 genes with distinct YC/YR usage to distinguish radiosensitive from resistant tumors that may have prognostic value. Finally, the authors argue that mTOR/PI3K inhibition, which has previously been shown to increase radiosensitivity, also increases YC usage in their organoid model, suggesting that these phenomena may be connected.

The main conclusion of this work is that YC/YR usage reflects a global transcriptional regulatory mechanism that correlates with cell states that are relevant to cancer growth and treatment. This is an exciting and novel concept that prompts many additional questions about the nature of the underlying mechanism. The impact of this message, however, is weakened by frequently confusing data presentation, unclear methodology, and some experimental concerns. This reviewer is enthusiastic about the main message of this manuscript and hope that the authors will be able to address these concerns. Details are provided below.

Major points:

1. A central finding is that YC usage increases in cancer cells categorized as “less differentiated”. A few clarifications are needed. First, the authors don’t say how they define differentiation status. Is this from the literature? Morphology? Expression of a gene signature? This should at least be described better in the methods.

Many thanks for highlighting the need for clarification on this. The profiling of differentiation status (alongside the other classification) was performed using publicly available profiling data about each of the tumour samples, as they are from well characterised cell lines. The source of this information were Cellosaurus (Expasy), Wikidata, and primary research sources referenced in Supplementary table 2. We have clarified the definition in the text at line 328-332.

Second, is it possible that proliferation rate is a stronger determinant of YC usage? For instance, Figure 2c shows that YC-enriched cancers have elevated expression of many cell cycle gene categories. The authors also hint at a relationship between proliferation, undifferentiated status, and radiosensitivity in the text (lines 392-394). The authors should test, at least for a subset of cancer cell lines in their panel, the relationship between proliferation rate and YC usage.

We thank the reviewer for this constructive suggestion. To address this, we have extracted the published doubling times of each of the cell lines used in this study (primarily from Cellosaurus). These have been added to Supplementary table 2 and a correlation plot of

doubling time to Mean Log₂FC YC:YR transcription (Cancer vs Healthy) has been generated (Extended data figure 1b). These analyses did not find a correlation between these factors (R=-0.26), however as now discussed in the manuscript text (line 380-381) this is not entirely surprising as, 1. published doubling times and cell line proliferation rates in practice often vary significantly with many factors contributing cell behaviour in culture. 2. The Mean Log₂FC YC:YR transcription (cancer vs healthy) metric is a product of the difference in behaviour between cancer and the matched healthy tissue samples. To truly match this, the doubling times also should be a metric of the relative proliferation rates the cancer and healthy tissues. However, the doubling rates of the healthy tissues is not available, so this comparison was not possible.

We have, however, conducted correlation analyses between organoid cell doubling times and YC transcript dynamic, where we do have control over the culturing conditions and found a strong correlation. This new data is included in and discussed in lines 581-593.

Third, do any mutations aside from p53 correlate with preferential YC usage? Mutation data for many of these cell lines is available from the Cancer Cell Line Encyclopaedia (Broad Institute).

This is again a very good suggestion. We have extracted the full mutational profiles of the cell lines used in this study where available (24/30 lines) and we identified mutations significantly associated with YC enriched and YC depleted cancers (Supplementary table 3 and text at lines 342-348).

2. Similar to the above comment, more information is needed about the CRC organoids described first in Figure 3. How were these obtained?

Details of how the organoids were maintained and experimentally assessed has been added to the Materials and methods section (subheadings, Organoid maintenance, doubling time assessment, radiotherapy and radio-sensitization assessment and imaging). Further details are available in the cited Wanigasooriya et al. 2022 paper (PMID: 35860583), which details how the organoids were initially generated from patient samples.

Do their proliferation rates correlate with IRR sensitivity?

We have calculated the doubling times of each of the organoid lines at the point where irradiation treatment starts and ends (4 and 9-days post seeding), with details of this analysis added to the Materials and methods, Figure 3h and Extended data figure 3i. This analysis revealed a clear association between YC usage, IRR sensitivity and proliferation rate.

What is their p53 mutation status?

The p53 mutational status of the organoids was detailed in Supplementary table 5, but this profiling has now been expanded to include mutation status from 20 different genes, where this was previously assessed in Wanigasooriya et al. 2022 (PMID: 35860583).

The authors showed striking differences in IRR sensitivity amongst the 5 CRC organoid lines. As with the cancer cell lines analyzed in Figure 2, is it possible that proliferation rate is be a better (and more quantitative) indicator of both YC usage and IRR sensitivity?

This is a very good point and indeed we do see that these three behavioural aspects are correlated. We have added discussion of this point in lines 581-585, 591-593 and 1131-1133.

3. In the overall TSS analysis of organoids in Figure 3f, it appears that YC usage generally changes while YR usage remains roughly constant. This also seems to be true for SND1 (Figure 3h). Are there examples where YR usage changes while YC usage remains constant? Or does only YC usage change? It would be helpful to visualize this, perhaps by providing a scatterplot showing levels of YC and YR TSSs in sensitive versus non-responsive organoids?

Indeed, varying degree of YR usage contributed to radiotherapy responsive YC:YR ratio change. This is demonstrated in a modified Extended data figure 3d, e where we now highlight the threshold of significant change in the scatter plot and we present in the revised d panel the number of genes with significant changes of YC and YR in a bar chart in e for better comparison. The description of these results is presented in lines 524-532.

4. Figure 4 shows that IRR decreases the YC/YR ratio in IRR-responsive organoids to levels similar to untreated non-responsive organoids. A simplistic interpretation would be that the decreased YC usage indicates a shift to a radio-resistant state, but this seems illogical. An alternative interpretation is that IRR selects for radio-resistant cells with low YC usage within organoids, and that the frequency of these cells is higher in resistant organoids. Could the authors comment on this possibility?

This is an excellent discussion point. YC:YR ratio change in response to irradiation could well be a result of selection for more radio-resistant tumour cells in heterogeneous organoids. We have expanded on this point in the discussion in lines 1160-1167, and added a panel to Figure 8 to illustrate this point.

5. The manuscript relies heavily on GO categories of genes to make the point that YC usage increases in many of the same genes under different contexts (e.g. differentiation status, irradiation) (Figure 2f, 3j, 4e). The meaning of these categories is confusing to this reviewer. Do they contain overlapping sets of genes? What is included in MTORC1 Signaling? Is it possible to instead define a unified gene signature of dual-initiator genes with variable YC/YR ratios from results in Figure 2f, and then analyze and/or refine this gene set in later figures? This would clarify the specific genes that link differentiation status to IRR sensitivity.

This is a good suggestion and has become a discussion point in the manuscript. Unfortunately, there is a relatively modest overlap between differentiation-associated and radiotherapy response-associated gene sets (10 genes shared in all) as assessed by gene identity. It was therefore not possible to define a unified gene signature of context-responsive dual initiators. However, we found that there were a number of shared

regulatory pathways associated with these gene sets, including PI3K/AKT/MTOR and MYC pathways which led to our further investigations of these. We have added new figure panels to Figure 6a,b to illustrate this together with the details of shared signalling pathway associated gene in a new Supplementary table 9. This is also discussed in lines 788-803.

The authors should also provide the list of the 148 radio-responsiveness signature genes shown in Figure 5a as a supplemental table.

As requested, we have listed the 4 gene sets with shared regulatory in Supplementary data 1-4s.

6. More information on the “delta-delta DE-Seq” analysis is needed. The approach is currently described only in the main text (lines 365-368), which appears to first normalize gene expression levels in cancer cells to levels in normal tissue. This is not appropriate for DESeq, which relies on raw read counts for the particular statistical model used to determine significance of differences in expression levels (Love et al., Genome Biol, PMID: 25516281). DESeq can handle ratios-of-ratios (e.g. YC versus YR for cancer versus normal) with the appropriate model design.

We thank the reviewer for these suggestions. We agree that DESeq is not the optimal pipeline for our purposes. Therefore, we have instead identified genes that are consistently enriched in YC enriched cancers over their matched healthy tissues, but unchanged or depleted in YC depleted cancers and vice-versa as described in lines 368-373, now shown in revised Figure 2c,d. This analysis identified a similar gene set and gene ontology profile for genes specifically upregulated in YC enriched cancers (cell cycle), and a slightly different gene set for genes specifically upregulated in YC depleted cancers, with their ontology reflective of cell migration, rather than metabolism. The key message remains unchanged however, that changes in YC transcript metabolism is reflective of cancer cells of different character or occupying distinct niches.

7. The authors use NVP-BEZ235 to test their hypothesis that PI3K/mTOR inhibition increases YC usage and radio-sensitivity. However, this molecule is also a strong inhibitor of ATM and ATR (Toledo et al., NSMB, PMID: 21552262), which may contribute more to the increased sensitivity to IRR than mTOR/PI3K inhibition. To show that this effect is specific to the mTOR and/or PI3K pathway, these experiments would need to be repeated with more specific inhibitors. Many such compounds are available.

As requested we have addressed this issue of specificity experimentally. We have repeated the radiosensitisation experiment detailed in Figure 7 by using the potent PI3K/mTOR dual inhibitor Omipalisib and the selective ATM/ATR inhibitor VE821. We note, that due to considerable crosstalk between the PI3K/Akt/mTOR and DNA damage response pathways, neither of these drugs are entirely specific to their respective pathways. Omipalisib has been shown to additionally inhibit DNA PK, although at a 10-fold lower specificity than PI3K and VE821 has been shown to downregulate mTOR at high concentrations (10 μ M), as a potential off target effect (Šalovská et al., 2018 - PMID: 30001349). Both drugs induced significant radio-sensitisation in the radio-resistant lines, similar to Dactolisib (see revised

Extended data figure 9a,b). Strikingly, in our transcript isoform specific RT-qPCR assay, both drugs also induced the same modulation of YC:YR ratios to Dactolisib. This being an enrichment in YC:YR ratios in the target genes upon treatment and restoration of the responsiveness of this ratio to irradiation (Extended data figure 9c). This observation suggests that YC transcript regulation is also affected by cellular stress induced by DNA damage, as is also suggested by the enrichment of DNA damage response associated genes in both the Radio-sensitivity trajectory and Irradiation-affected trajectory gene sets. We have discussed these results in lines 962-979 and changed the subheadings (lines 910-911). We also changed the ms title to better reflect the broader range of proliferation-associated signaling pathways impacting on YC:YR dynamics.

Furthermore, the increase in YC transcript levels could also be explained by increased mRNA stability following BEZ235 treatment, since mTOR inhibition stabilizes YC mRNAs. The authors should also at least discuss the possibility that the increase in YC usage in inhibitor-treated cells may reflect changes in stability rather than transcription (and preferably show this directly using reporters).

We agree with the reviewer that the effect of both NVP-BE235 and Omipalisib on YC usage is likely through a change in the metabolism of YC transcripts by either/both a change in transcript stability and production and we explore this further in the discussion (line 1052-1054). It is currently beyond the tools available to us to reliably disentangle the relative contributions of transcriptional and post-transcriptional regulation due to the difficulty of distinguishing YC and YR origin of transcripts by reporter assays. Therefore we suggest that the development transcriptional and posttranscriptional contribution analyses will be subject to future research. We specifically discuss these challenges in the Discussion section (lines 1055-1060).

8. The manuscript includes a lengthy discussion (lines 626-683) of whether TOP mRNAs, which could be considered extreme examples of YC-hi genes, drive the increase in YC usage in radio-sensitive samples. The conclusion is that increased TSS usage for TOP genes reflects a broader phenomenon that globally increases YC usage. As nearly all of the associated data is in the Supplemental section, it seems that this section could be significantly condensed. As written, it distracts from the main message that YC usage is globally increased through a mechanism that is not yet understood.

Upon careful consideration of the potential importance of the relationship between YC initiation and 5'TOP mRNA production and metabolism and in response to the related comments of reviewer #2 we opted to elevate parts of this section into the main figure of the manuscript. TOP mRNA regulation has been well established and has been given considerable focus both by the RNA biology and cancer research fields. Our results with dual initiation promoters in cancers suggest that TOP mRNAs are produced from a far larger set of genes than previously appreciated. This finding opens up new research directions to explore the biologically relevant sequence determinants of TOP mRNA metabolism and the interplay between transcription initiation and posttranscriptional regulation acting on the very same 5' end sequences. For these reasons, we decided to give more prominence to these data by moving the key data previously presented in Extended figure 5 to a new main Figure 5. In addition, we have pointed out the potential importance of YC initiation in dual

initiation genes as a previously unappreciated potential TOP mRNA production mechanism in the Discussion and in the revised abstract.

Minor points:

1. Figures 1b and 1c are conceptually redundant. Is it possible to combine them into a single panel?

We have combined these figures into Figure 1b

2. How many tumor samples were analyzed in Figures 2a-d? 6 responsive and 6 non-responsive are listed in Supplemental Table 4, but Supplemental Table 5 suggests that only one from each category was analyzed by CAGE. Is that correct? More information is needed in the Methods section as to how these samples were obtained.

CAGE was performed on 8 CRC primary FFPE tumour samples (4 responsive and 4 non-responsive). To simplify analysis, the data from these samples was pooled and subjected to the analyses shown in Figure 3a-c. 12 CRC tumour samples were used for the qPCR validation shown in Figure 6e (the 8 CAGE ones plus additional responsive and non-responsive samples). This information is listed in Supplementary table 5, but the reviewer is right to point out that this detail is not as clear as it should be in the main text. We have therefore added further explanation of this in lines 452-454 and 886-891. Details about how the samples were obtained can also be found in line 126-131.

3. The heatmaps shown in Figures 2e, 3i, 4d seem unnecessary, as these gene sets have been selected based on a shared pattern of expression. I recommend that they be removed.

These have now moved to Extended data figures 1, 3 and 4. We have kept them available to aid illustration of how these gene sets were selected.

4. This sentence at lines 456-458 is unclear: “Additionally, the ability of the technique to identify dual initiator promoters appears to be considerably poorer than traditional CAGE used for the FANTOM5 cancer data (186 Dis [5.3% of promoters, FFPEcap-seq], 3475 Dis [19.9%]).” What is “Dis”? 3475 is 19.9% of what?

We apologise for the lack of explanation. We have now revised this abbreviation to DIP (dual initiation promoter) and explain it and the numbers better in the revised ms: *“Additionally, the ability of the technique to identify dual initiator promoters (DIP) appears to be considerably poorer than traditional CAGE used for the FANTOM5 cancer data (186 DIP [5.3% of all promoters] in FFPEcap-seq, versus 3475 Dis [19.9% of all promoters] in the FANTOM5 CAGE-seq data).”*

5. Is the increased expression of YR transcripts in the “other promoter” category for CRC4 & 5 (Figure 3f) significant?

Yes, the dynamics of YC and YR in non-dual promoters is also significant (Figure 3e). This suggests a broader switch in YC vs YR usage between the organoid lines, beyond just dual initiators, which we comment on in the main text at lines 514-516.

6. It would make more sense to show the qPCR analysis of clinical CRC samples and the analysis of CRC organoid samples as separate figure panels. The analysis of CRC organoids validates the ability of the qPCR method to distinguish YR and YC usage for these transcripts, but the correlation with radio-sensitivity is expected because these genes were identified based on results from these same samples. The analysis of clinical CRC samples is much more important because it validates the strategy for assessing samples that were not used to identify this gene signature. Moving these results to their own panel would strengthen this point.

We thank the reviewer for this suggestion. We have now split Figure 6d into revised Figure 6d,e panels to address this and added emphasis in the text of the differing roles of the two panels for validation and for demonstrating the prognostic value of YC:YR transcript dynamics by CAGE-seq assay.

Reviewer #2:

Remarks to the Author:

Wragg et al. report that the transcription initiation site usage switches from YR to YC in cancer samples and this switch is correlated with differentiation status of cancer types. They identified the enrichment of YC usage in dual initiators is related to radiotherapy responsive CRC and CRC organoid samples. By inhibiting the PI3K/Akt/mTOR pathway, CRC organoids insensitive to radiotherapy was shown to restore sensitivity. Overall, the promoter switch is a consequence of mTOR alternation which is measured by CAGE. The method and conclusion of the study are appropriate, and the content of this study is worth publishing. However, I have some questions and comments as follows:

1. Do all the CRC organoids show cancer phenotypes? I assume it can be extracted from the organoid CAGE data or other IF results easily.

Yes, each of these organoids do show cancer phenotypes, closely matching the character of their tumour of origin. This is published in Wanigasooriya et al. 2022 (PMID: 35860583), which describes the original generation and characterisation of these organoids. The reviewer is right to suggest that this should be highlighted in this manuscript. To this effect we have added phenotypical analysis of organoid used in this study (proliferation rate, mutational profile, morphological features) in revised Figure 3h, Supplementary table 5 and Extended data figure 3i.

According to the first part of the study, lower YC/YR ratio may refer to more differentiated cancer or non-cancer. If radiotherapy select to kill more aggressive (undifferentiated) types but not the non-cancer cells, is it needed to manipulate the cells to be killed? On the other hand, if the Dacto treatment is applied to patients, will non-cancer cells be sensitized to radiotherapy? These should be discussed in the manuscript.

These are great suggestions for discussion points which have now been added to the Discussion in lines 1128-1143.

If a healthy organoid is available, same experiment described in Fig. 6 will directly resolve this concern.

Regarding the suggestion to directly test the safety profile of Dactolisib radiosensitisation in healthy colon organoids, this is a very interesting suggestion. Unfortunately, the healthy organoid lines we have available are not routinely used and have failed to thrive, despite trying several set of conditions on three separate cultures. As this analysis, though important, is a perhaps more tangential to the focus of the paper, and we have not pursued this line of work further.

2. If data is available, it will be good to include non-cancer clinical sample / organoid from healthy subject in Fig. 3B & F, to observe if there is any YC/YR difference between non-responsive and normal.

This is a great suggestion, but as described in the previous answer none of our non-cancer organoids thrived in culture, so we were unable to get a reliable YC/YR ratio for these

cultures, reflective of their healthy state, to augment the cancer vs. healthy comparisons detail in Figures 1 and 2.

3. What are the selection criteria for the 4 genes with qPCR results?

The candidate genes for RT-qPCR analysis were selected from the 147 radio-responsiveness signature gene detailed in Figure 5. They were further selected on the basis of having sufficient separation of the YR and YC sites of expression within the promoter that RT-qPCR primers could be designed that would only amplify one but not the other, so that the relative transcription from YC and YR sites could be assessed by RT-qPCR. We have added further explanation in the text to clarify this. We have also highlighted these genes in a YC:YR Log3FC scatter plot in Extended data figure 3d, to demonstrate their representative nature.

For the least responsive gene, GMPR2, is it free of related TF binding motif?

This is a very interesting question. We have assessed the presence of the TF binding motifs in the 4 qPCR candidate genes including GMPR2 in Extended data figure 7e. For GMPR2 in particular, this analysis revealed no ELK1, GABPA, MYC or TP53 TF binding sites in the YR predominant section of the promoter, but 2 potential GABPA binding location (on the basis of motif) proximal to the YC initiation site.

It would be interesting to see the CAGE-seq result derived from the Dacto-treated organoid to reveal the actual effect on specific dual initiators.

This is a very good suggestion. We have added a genome browser view of the dual initiator C9orf85 to Figure 7d to illustrate the effect of Dactolisib on YC:YR transcript dynamics.

4. In Fig. 2F and Fig. 5B, non-significant pathways were cherry-picked to show. I don't think it is necessary and meaningful. And in line 699: "MYC, MTORC1 and P53 regulatory pathways (Figure 5B, Extended data figure 5A), in agreement with Figure 2F, Figure 3J and Figure 4E." MYC is significant in 3 out of the 4, MTORC1 and P53 is significant in 2 out of the 4. Please revise this statement.

The reviewer is correct in pointing out that some of the pathways shown in Figure 2F (now Extended data figure 1e) and Figure 5b (now Extended figure 6b) are on the borderline of significance (0.03-0.08 FDR). These have not been cherry picked as they indeed reflect the top pathways associated with gene sets. Following this reviewer's suggestion we have amended the discussion of these figure panels to reflect the borderline nature of their significance (lines 396, **797** and **812**). We have also refocused our illustration of shared regulatory pathways in a revised Figure 6b panel to better reflect the significance levels of pathways in each gene set.

Alternatively, test level analysis can be performed across the pathways to evaluate if MYC, MTORC1 and P53 are consistently enriched significantly in the 4 tests (10.1038/s41467-019-13983-9)

Many thanks for this suggestion. We used the Active Pathways pipeline, which requires gene sets with p-values to allow ranking of the genes. The trajectory analyses presented in the paper do not allow this as they contain too many variables for the De-seq data frame.

We generated simplified reproductions of the analyses with appropriate p-values using ratio of ratio analyses in de-seq (YC vs YR, Responsive vs. Non-responsive or Irradiated vs non-irradiated— where YC and YR are treated as separate alleles of the same gene). We did not take into account moderately responsive tumours or genes that show differential YC:YR response upon irradiation, between responsive and non-responsive cohorts.

Similarly, in the pan-cancer analysis De-seq is unable to deal with comparing all three of YC vs YR, Cancer vs Healthy, YC enriched vs YC depleted. Instead, we ran the active pathways pipeline on the simplified comparators listed above (YC vs YR, Responsive vs. Non-responsive and YC vs YR, Responsive IRR vs Control). The results (Reviewer response figure 1) were similar to the GOs in the trajectory analyses. It looks most similar to the Responsive vs. Non-responsive analysis shown in Extended data figure 3h, probably because this was the only analysis in the data frame we were able to (close to) fully replicate. We struggled to integrate this data without complicating an already quite complex manuscript without adding to its message. As the active pathways analysis incompletely reflects the analyses shown in the manuscript, yet supports a similar conclusion, we decided to not include this data, but rather to reorganise our existing gene ontology analyses, to better reflect the intersection of pathway associations between gene set in the revised Figure 6b.

5. Downstream 5'TOP-like can be tested directly (line 662) from the YC-other. It would be good if the authors can perform the analysis and reveal this possibility.

This is a very good suggestion. We have now added an analysis of internal TOP transcripts (transcripts with a 5+ polypyrimidine stretch in the first 50 bp) initiating with both YC and YR dinucleotides to Extended data figure 5c,d. This analysis revealed that the presence of an internal TOP motif did not add to the radio-responsive dynamics of either YC or YR transcripts.

6. For the TF binding motif analysis, please include available motif in JASPAR.

We have added these details to Extended data figure 7 and the relevant Materials and methods section.

Reviewer #3:

Remarks to the Author:

Variations in transcription start site (TSS) selection reflect the diversity of preinitiation complexes and can impact post-transcriptional RNA fates, especially the recent identification of dual-initiation promoters genes carrying canonical initiation with pyrimidine/purine (YR) dinucleotide while carrying polypyrimidine initiator (5'-TOP) translation machinery-associated genes. However, the functional significance of start site choice in these dual promoter architectures is little understood. The authors first systematically identified genes that mark YC enriched vs YC depleted in cancers. Enriched YC transcription correlated with poorly differentiated, proliferative (and potentially TP53 mutated) cancer subtypes led to a more specific examination of the developmental regulation of TSS selection in the chemo radio-resistance of the CRC.

Overall, the hypothesis is novel, and the data also propose for the first time that dual initiation on shared promoters represents a composite promoter architecture in genes involved in cancer biology and diversity in treatment.

The manuscript is also written well, and however, I have some mid & minor concerns before considering for publication.

Mid/ Majors:

1. The author could construct a reporter vector with dual-initiation sequences/ promoter fused to luciferase (LacZ gene) and evaluate some candidate genes correlated with different stages of patient samples in CRC cell lines.

This is a really nice suggestion. The generation of a YC:YR dual initiation reporter system to look at transcript temporal dynamics is definitely something we are looking to do in the future. However, given the scale and range of experimental approaches already presented on this dense manuscript we were concerned that the reporter assays, which would need validation before subsequent characterisation of YC:YR transcript dynamics at different stages of organoid development and treatment regimens would be better suited to a future paper. We are also concerned about the technical challenges, we have not yet resolved, associated with the need to separate reporter signals emanating from YC or YR component of dual initiation promoters without interfering the potential overlap in their unknown sequence determinants.

2. As radiation therapy is more associated with rectal cancer, it is beneficial for authors to define which of CRC 1-5 are rectal or colon-derived organoids.

We thank the reviewer for their suggestion and renamed the criticised datasets from upper, mid and lower (Supplementary table 5) to colon, sigmoid and rectal tumour sites. Twelve of the 18 CRC tumour samples used in this study are of rectal origin, with 3 colon and 2 sigmoid.

3. The authors need to describe in more detail how and why they used 25Gys of irradiation over five days and their genetic background.

The 5x5Gys (25 Gys) treatment regimen was designed to replicate the standard short course treatment used in the clinic. We have added further description of this in the text at lines 493-494.

Regarding genetic background, details of the p53, KRAS and NRAS mutational status of the organoids was detailed in Supplementary table 5, but this profiling has now been expanded to include mutation status from 20 different genes where this was previously assessed in Wanigasooriya et al. 2022 (PMID: 35860583).

4. Also, authors could provide images of organoids from responsive, moderately responsive and non-responsive for their phenotypical characteristics.

This is a great suggestion and helps in understanding of how different colorectal cancer states, reflected by distinct organoid morphology might be associated with radio-sensitivity and YC metabolism. To address this, we have added brightfield images of each of the organoid lines Figure 3j with phenotypical commentary added in the text. Strikingly, the radio-resistant organoids both had cryptic organoid morphology (unlike the others), which is reflective of a more differentiated (normal intestine like identity), which aligns well with the pan-cancer analysis linking differentiation with relative YC depletion.

Minors;

1. In figure 6a, the organoid images are the same as each other

Given the density of the figures, we opted to remove this treatment strategy schematic, instead expanding the description of this strategy in the Material and methods section under the Subheading "*Organoid radiotherapy and radio-sensitization assessment*".

2. If figure 6b, p-values are missing

We have now been added the missing P-values.

3. A couple of sentences need a bit of rewriting, e.g., on Page 16, line 427- suggesting efficient start site detection and the that biological replicates of responsive

We thank the reviewer for these editing suggestions. We have rephrased the sentence to: "CAGE reads were mapped to the human genome assembly (GRCh38/hg38) and CTSSs assigned. ~55-60% of CTSSs mapped to the promoter region of genes. This is in line with the previously demonstrated specificity of the FFPEcap-seq approach²⁸ (Extended data figure 2a, Supplementary table 6) and demonstrated efficient and reproducible start site detection. Biological replicates of responsive and non-responsive CRC tumours were merged to ask whether YC transcription correlates with CRT responsiveness."

References-

Wanigasooriya, K., et al. Patient Derived Organoids Confirm That PI3K/AKT Signalling Is an Escape Pathway for Radioresistance and a Target for Therapy in Rectal Cancer. *Frontiers in oncology* **12**, 920444 (2022).

Šalovská, B., et al. Radio-sensitizing effects of VE-821 and beyond: Distinct phosphoproteomic and metabolomic changes after ATR inhibition in irradiated MOLT-4 cells. *PLoS One* **13**, e0199349 (2018).

Decision Letter, first revision:

Message: Our ref: NSMB-A47311A

12th Sep 2023

Dear Dr. Wragg,

Thank you for submitting your revised manuscript "Transcription initiation site choice precedes proliferation signalling-dependent RNA metabolism and defines cancer differentiation and radio-sensitivity" (NSMB-A47311A). It has now been seen by the original referees and their comments are below. The reviewers find that the paper has improved in revision, and therefore we'll be happy to accept it in principle in Nature Structural & Molecular Biology, pending minor revisions to satisfy the referees' final requests and to comply with our editorial and formatting guidelines.

We are now performing detailed checks on your paper and will send you a checklist detailing our editorial and formatting requirements in about two weeks. Please do not upload the final materials and make any revisions until you receive this additional information from us.

To facilitate our work at this stage, it is important that we have a copy of the main text as a word file. If you could please send along a word version of this file as soon as possible, we would greatly appreciate it; please make sure to copy the NSMB account (cc'ed above).

Sincerely,

Dimitris Typas
Associate Editor
Nature Structural & Molecular Biology
ORCID: 0000-0002-8737-1319

Reviewer #1 (Remarks to the Author):

The authors have addressed my concerns and I recommend this thought-provoking manuscript for publication.

Reviewer #2 (Remarks to the Author):

The authors sufficiently addressed all comments raised by this reviewer.

Reviewer #3 (Remarks to the Author):

The revised version of the manuscript has significantly enhanced the clarity and quality of the pertinent presentation of the data and conclusions, particularly in the discussion section. The authors have addressed my concerns with a relatively comprehensible response within the specified deadline, which is acceptable.

Final Decision Letter:

Message 19th Oct 2023

:

Dear Dr. Wragg,

We are now happy to accept your revised paper "Intra-promoter switch of transcription initiation sites in proliferation signalling-dependent RNA metabolism" for publication as an Article in Nature Structural & Molecular Biology.

As soon as your article is published, you can generate your shareable link by entering the DOI of your article here: http://authors.springernature.com/share. Corresponding authors will also receive an automated email with the shareable link

Your paper will be published online soon after we receive proof corrections and will appear in print in the next available issue. You can find out your date of online publication by contacting the production team shortly after sending your proof corrections. Content is published online weekly on Mondays and Thursdays, and the embargo is set at 16:00 London time (GMT)/11:00 am US Eastern time (EST) on the day of publication. Now is the time to inform your Public Relations or Press Office about your paper, as they might be interested in promoting its publication. This will allow them time to prepare an accurate and satisfactory press release. Include your manuscript tracking number (NSMB-A47311B) and our journal name, which they will need when they contact our press office.

About one week before your paper is published online, we shall be distributing a press release to news organizations worldwide, which may very well include details of your work. We are happy for your institution or funding agency to prepare its own press release, but it must mention the embargo date and Nature Structural & Molecular Biology. If you or your Press Office have any enquiries in the meantime, please contact press@nature.com.

Please note that *Nature Structural & Molecular Biology* is a Transformative Journal (TJ). Authors may publish their research with us through the traditional subscription access route or make their paper immediately open access through payment of an article-processing charge (APC). Authors will not be required to make a final decision about access to their article until it has been accepted. Find out more about Transformative Journals <https://www.springernature.com/gp/open-research/transformative-journals>

Authors may need to take specific actions to achieve [open access](https://www.springernature.com/gp/open-research/funding/policy-)

compliance-faqs"> compliance with funder and institutional open access mandates.

If your research is supported by a funder that requires immediate open access (e.g. according to [Plan S principles](https://www.springernature.com/gp/open-research/plan-s-compliance)) then you should select the gold OA route, and we will direct you to the compliant route where possible. For authors selecting the subscription publication route, the journal's standard licensing terms will need to be accepted, including [self-archiving policies](https://www.springernature.com/gp/open-research/policies/journal-policies). Those licensing terms will supersede any other terms that the author or any third party may assert apply to any version of the manuscript.

Sincerely,

Dimitris Typas
Associate Editor
Nature Structural & Molecular Biology
ORCID: 0000-0002-8737-1319

Click here if you would like to recommend Nature Structural & Molecular Biology to your librarian:

<http://www.nature.com/subscriptions/recommend.html#forms>